# Development of Phytochemical Delivery Systems by Nano-Suspension and Nano-Emulsion Techniques

**DOI:** 10.3390/ijms24129824

**Published:** 2023-06-06

**Authors:** Guendalina Zuccari, Silvana Alfei

**Affiliations:** Department of Pharmacy (DiFAR), University of Genoa, Viale Cembrano 4, I-16148 Genova, Italy

**Keywords:** nanotechnology application, nanosuspension techniques, nanoemulsion techniques, bioactive constituents of plant, phytochemicals (PHYs), poor water solubility, drug delivery systems (DDSs), toxicological risks of NPs, acute and chronic nanotoxicity, clinical applications

## Abstract

The awareness of the existence of plant bioactive compounds, namely, phytochemicals (PHYs), with health properties is progressively expanding. Therefore, their massive introduction in the normal diet and in food supplements and their use as natural therapeutics to treat several diseases are increasingly emphasized by several sectors. In particular, most PHYs possessing antifungal, antiviral, anti-inflammatory, antibacterial, antiulcer, anti-cholesterol, hypoglycemic, immunomodulatory, and antioxidant properties have been isolated from plants. Additionally, their secondary modification with new functionalities to further improve their intrinsic beneficial effects has been extensively investigated. Unfortunately, although the idea of exploiting PHYs as therapeutics is amazing, its realization is far from simple, and the possibility of employing them as efficient clinically administrable drugs is almost utopic. Most PHYs are insoluble in water, and, especially when introduced orally, they hardly manage to pass through physiological barriers and scarcely reach the site of action in therapeutic concentrations. Their degradation by enzymatic and microbial digestion, as well as their rapid metabolism and excretion, strongly limits their in vivo activity. To overcome these drawbacks, several nanotechnological approaches have been used, and many nanosized PHY-loaded delivery systems have been developed. This paper, by reporting various case studies, reviews the foremost nanosuspension- and nanoemulsion-based techniques developed for formulating the most relevant PHYs into more bioavailable nanoparticles (NPs) that are suitable or promising for clinical application, mainly by oral administration. In addition, the acute and chronic toxic effects due to exposure to NPs reported so far, the possible nanotoxicity that could result from their massive employment, and ongoing actions to improve knowledge in this field are discussed. The state of the art concerning the actual clinical application of both PHYs and the nanotechnologically engineered PHYs is also reviewed.

## 1. Introduction

Epidemiological studies have evidenced that the ingestion of some foods, including edible plants, is associated with the onset of beneficial health effects. An example is the assumption that red wine is correlated with a decrease in death from cardiovascular events caused by atherothrombosis due to its capability of decreasing the progression of atherosclerotic lesions [1]. Additionally, it has been demonstrated that green vegetables exert protective effects against cardiac diseases [1]. The bioactive chemical compounds responsible for these benefits are known as phytochemicals (PHYs). Specifically, PHYs are defined as bioactive chemical compounds found in plants, such as fruits, vegetables, grains, and other plant-derived foods, that may supply health benefits beyond basic nutrition and could help to reduce the risk of major chronic diseases [2]. PHYs are generally produced by plants to help themselves resist fungi, bacteria, and plant virus infections and also to hamper their consumption by insects and other animals [3]. Over the years, humans have used PHYs both as poisons and as traditional medicine [3]. Nowadays, the recognition that plants can be a source of compounds endowed with beneficial properties is progressively expanding worldwide, and both the food market and the sector of natural compounds are involved [1]. Moreover, experts in the field incessantly emphasize their extensive introduction in the normal diet and in food supplements, as well as their use as natural therapeutics to treat several diseases [4]. As of the beginning of January 2022, a total of 130 thousand PHYs have been found [5], and others will be discovered in the next several years. Several PHYs with many beneficial effects, including antifungal, antiviral, anti-inflammatory, antibacterial, antiulcer, anti-cholesterol, hypoglycemic, immunomodulatory, and antioxidant activities, have been isolated from plants, such as vegetables, herbs, fruits, legumes, oils, spices, nuts, and whole grains [6,7,8,9,10,11]. Additionally, scientists, engineers, and technologists continually investigate the possibilities of the chemical modification of known PHYs by introducing new antioxidant, anti-free-radical, and anticancer functionalities to boost their intrinsic activities. Unfortunately, although the idea of exploiting the strong potential of PHYs for health purposes is brilliant, its realization is far from simple [12]. Additionally, the thought of using extracted PHYs as effective orally or otherwise administrable health promoters is almost utopic. In the case of oral administration, solubility in an aqueous medium, permeability across the membrane of intestinal epithelial cells, and molecular interactions in the fluids of the gastrointestinal tract (GIT) are key factors that greatly affect the possible beneficial effects of PHYs by influencing their route to the bloodstream and their final distribution to the target. In this regard, most PHYs are insoluble in water, and, in whatever form they are introduced, especially if orally administered, they hardly manage to pass through physiological barriers, thus reaching the target only at sub-therapeutic concentrations. Moreover, their fast degradation by chemical, enzymatic, and microbial digestion, which can occur in the mouth, stomach, and small and large intestines [12], as well as their rapid metabolism and excretion via the kidney, biliary system, or lung, further restrict their in vivo activity [13]. Collectively, PHYs commonly reach only nano/picomolar concentrations in cells and tissues, amounts that are insufficient to produce a therapeutic response. Concerning oral administration, Figure 1 shows the main possible drawbacks related to most PHYs and the events that limit their in vivo beneficial effects after administration.

Additionally, since, when ingested, PHYs can be altered by microbial fermentation, their biological properties may also be distorted and paradoxically converted into toxic effects that were not observed in in vitro tests. To overcome the aforementioned problems and to allow the exploitation of PHYs as health enhancers, researchers increasingly resort to nanotechnology and nanostructures with dimensions of nanometers (nm). Over the years, several PHY-enriched nanomaterials have been engineered to overcome the poor solubility, permeability, and negative pharmacokinetics of PHYs, and different nanosized delivery systems (DSs) to transport therapeutic concentrations of PHYs to their targets have been designed [12,14]. Moreover, aiming at the better treatment of diseases, such as cancer, using PHYs, novel nanotechnological carriers for the codelivery of PHYs and conventional anticancer drugs have been developed with various benefits, including improved solubility, reduced adverse effects, higher efficacy, reduced dose, improved dosing frequency, reduced drug resistance, improved bioavailability, and higher patient compliance [15]. In this context, an enzymatically transformable polymer-based nanotherapeutic approach to eliminate minimal relapsable cancer was proposed by Li et al. [16]. In particular, it exploits matrix metalloproteinase (MMP) overactivation in cancer tissues to guide the codelivery of colchicine, a microtubule-disrupting and anti-inflammatory alkaloid originally extracted from plants of the genus *Colchicum* (especially *Colchicum autumnale*) and marimastat (an MMP inhibitor) [16].

Figure 2 below provides an idea of the growing scientific interest in using PHYs for health purposes and of the consequent application of nanotechnologies for solving their physicochemical issues during the last 15 years. In particular, the bar graph in Figure 2 was obtained by carrying out a survey about the number of works published on the topics from 2008 to date (excluding the current year), performing a search by using the keywords “nanomaterials OR nanoparticles OR nanoformulation AND phytochemicals”.

While the interest in the topic was low up to the year 2015 (fewer than 100 papers), in the following years, it grew exponentially, as established by the several papers that have been published. By reporting several case studies, this review provides an up-to-date overview of nanotechnological approaches based on nanosuspension (NS) and nanoemulsion (NE) techniques developed so far for formulating the most relevant PHYs in more bioavailable NPs, promising for clinical applications, especially focusing on those that are orally administrable. We also pay attention to the pending issue relating to the possible toxic effects of NPs on humans, animals, and the environment in a scenario of limited knowledge, and we briefly report the ongoing actions to improve expertise in the field and/or to promote the development of increasingly safe nanomaterials.

The last section of the present paper provides a glimpse of the actual clinical applicability of PHYs and of nanotechnologically engineered PHYs by reviewing those that are already clinically approved or are currently in advanced clinical trials. As an original contribution, most information provided here is also graphically presented and has been statistically analyzed.

## 2. Phytochemicals (PHYs)

### 2.1. Phytochemicals and Nutraceuticals: Not Quite the Same

Both PHYs, also known as phytonutrients, and nutraceuticals are bioactive compounds that can be found in edible products possessing beneficial properties capable of enhancing human health and are commonly ingested with the diet. Anyway, while nutraceuticals are essential nutrients for human life and are necessarily present in edible products derived from animals, plants, or fungi, PHYs are non-nutrients and exclusively originate from plants [17]. Although a lower intake of PHYs does not cause defects, these compounds can be invaluable to human health, and a diet rich in PHYs is strongly connected to better health [18]. In vitro studies have demonstrated that PHYs are multifunctional compounds with health-promoting properties like those of conventional drugs and can be considered “pharmaceutical-grade compounds”. They can improve the ability to detox, boost the immune system, and help to protect against age-related diseases, such as diabetes, heart disease, and osteoporosis [18].

### 2.2. Phytochemicals: An Overview

It has been reported that scientists have already identified over 5000 different classes of PHYs. Anyway, many more remain undiscovered, and much more has to be learned about their potential benefits [19]. Although PHYs can derive from both edible and non-edible plants [20], all plant-based foods, including fruits, vegetables, nuts, and herbs, contain them. Generally, PHYs confer to the plant-derived food containing them a particular bright color, but foods without these characteristics can contain these healthy promoters, as well (Table 1). Green, purple, red, blue, or yellow vegetables and fruits contain colored PHYs [19,21,22], but non-brightly colored potatoes, cauliflower, nuts such as almonds, cashews, and hazelnuts, tea, and dark chocolate also contain several PHYs (Table 1) [19,23,24,25]. Moreover, essential oils (EOs), including those from pine needles, cedar, and lavender, are used as health promoters due to their PHY contents (Table 1) [26,27,28].

#### 2.2.1. Specific Sources and Benefits of the Main Types of PHYs

The most common PHYs are polyphenols, carotenoids, coumarins, indoles, organosulfur compounds, isothiocyanates, saponins, tannins, phenylpropanoids, anthraquinones, ginsenosides, terpenoids, etc. [39]. In Table 2 below, we report the most relevant PHYs, their sources, and the associated health-promoting effects [3,40].

As mentioned above, the known PHYs are in the thousands. In Figure 3 and Figure 4, we report the chemical structures of the most common polyphenols (Figure 3) and of other relevant PHYs (Figure 4) with their properties [1,3,10,11,17,18,50,51,52,53,54,55,56,57,58,59,60,61,62,63,64,65].

#### 2.2.2. Let Us Eat in Color

Since PHYs confer particular food colors, colored foods surely contain PHYs and possess the same benefits as PHYs having the same color. So, by eating foods from all of the different color groups, a wide range of different PHYs can be assumed, with a consequent broad spectrum of benefits [66]. Table 3 below reports the general subdivision of plant-related foods into the different color groups.

## 3. Physicochemical and Pharmacokinetic Drawbacks Limiting the Development of Phytochemicals as Clinically Administrable Therapeutics

The development of PHYs as clinically administrable therapeutics by different routes, including oral, rectal, transdermal, or subcutaneous ones, is strongly restrained by several physicochemical and pharmacokinetic limitations, mainly including low water solubility, poor bioavailability, and deficient targeting [67]. Concerning oral administration, the dissolution of a poorly water-soluble compound in the intestinal fluids, as well as its GIT permeation, is very slow, and its systemic concentration will hardly be enough to produce a significant therapeutic response [1]. Poor water solubility implies a low absorption rate at the gut level, low bioavailability, and insufficient blood and tissue concentrations. Consequently, although when tested in vitro, PHYs show a plethora of beneficial effects and high activity, they are almost ineffective when assayed in vivo. Figure 5 reports the main factors that can influence the water solubility and, therefore, the in vivo bioavailability of a bioactive compound.

In particular, unsymmetrical small-size particles with high surface area dissolve better and more quickly. Higher temperatures promote dissolution, while a high molecular weight (MW) lowers compounds’ solubility. Since it is known that branched polymers are more soluble than linear ones with equal MW, a large amount of branching in carbon chains favors solubility. In fact, branched-chain molecules have low volume/dimension ratios in solutions and higher dissolution rates. Molecules arranged in amorphous forms have higher aqueous solubility than crystalline ones, and different polymorphs have different solubilities. Additionally, since ionized forms have a higher solubility in water and weak acids or weak bases ionize in solution on the basis of the pH of the medium, the pH of solutions can strongly affect a compound’s solubility. An approach to improve the water dispersibility of bioactive compounds could consist of formulating them in colloidal suspensions or emulsions, but when their solubility is excessively low, very high concentrations of surfactants, stabilizers, polymers, osmotic agents, organic solvents, or complexing agents will be required, which may trigger unpleasant side effects, including GIT irritation, in future oral administration [68,69]. In this context, efficient and low-cost solubilizing methods that minimize or even avoid the use of harmful excipients, such as organic solvents, cosolvents, emulsifiers, or other additives, are necessary. Currently, nanotechnology is extensively exploited to improve the solubility of bioactive compounds, including PHYs, to help their formulation and to enhance their absorption and bioavailability, with the final goal of closing the gap between the exciting and very promising results obtained in vitro and the unsatisfying results obtained in in vivo studies [70,71,72,73].

### 3.1. Improving Solubility of Bioactive Compounds

Bioactive compounds’ solubility can be improved by two modalities, which are both based on the reduction in particle size to micrometer or nanometer dimensions.

#### 3.1.1. Particle Engineering Techniques (PETs)

Particle engineering techniques (PETs) consist of old and novel methods to improve the solubility and bioavailability of a compound. By using PETs, the physicochemical, micromeritic, and biopharmaceutical properties of a compound are changed, mainly by reducing its particle size [74]. In addition to wet milling, dry milling, high-pressure homogenization (HPH), and ultra-high-pressure homogenization (UHPH), novel PETs comprise supercritical fluid technologies and cryogenic technologies [74,75,76]. The techniques belonging to these PETs are reported in Figure 6.

#### 3.1.2. Formulation Approaches (FAs)

FAs aim to obtain solid, lipid, or amorphous nanoformulations from colloidal dispersions, which in turn are prepared using mixtures of water/oil phases, stabilizers, and solvents/cosolvents by using PETs, such as spray drying (SD), milling, and other techniques reported in Figure 6. Table 4 collects the most commonly used techniques with their related advantages and disadvantages.

Except for SD, SFD, SFL, and TFF, all the techniques in Table 4 have the disadvantages of retaining a lot of the residual solvent and yielding low percentages of EE%. Additionally, except for cryogenic methods, they can cause a reduction in the biological activity of bioactive compounds due to thermal degradation or other undesirable events. All are multi-stage processes, frequently requiring an additional micronization step by air jet milling to obtain the needed particle size and size distribution, which could cause occasional crystallographic defects in the products [17]. In contrast, although costly, the SFE method is a single-step process requiring a shorter operation time. In the SFE method, the residual solvent content can be monitored, and micronized dry powders with controllable particle size, morphology, and crystallinity can be achieved [17].

## 4. Nanotechnology

Currently, nanotechnology is the most promising science, engineering, and technology conducted at the nanoscale (1–100 nm) and is used to improve the solubility of bioactive compounds in GIT fluids, their GIT permeability, and their bioavailability. The nanomanipulation of drugs can allow them to more easily bypass the physiological barriers that limit their oral delivery. As recently reported, especially if strongly based not only on elaborate designs and chemical–physical strategies but also on the thorough dissection of the mechanisms to obtain reproducible results, synthetic nanomedicine in the future will have nonpareil advantages in drug delivery, as well as in clinical practice [78].

### 4.1. Physiological Barriers to Oral Drug Delivery

The oral absorption of a bioactive compound is the process by which it is transported from the GIT to the blood compartment. Several factors affect oral absorption, including poor aqueous solubility and therefore a slow dissolution rate in GIT fluids, instability in the acidic environment of the stomach, the presence of degrading enzymes in the GIT, the presence of food, biological barriers, and finally, first-pass metabolism in the liver [79]. The GIT encompasses the oral cavity, esophagus, stomach, small intestine, and colon. Each region of the GIT has different characteristics that affect drug uptake, which can be hindered by complex physiological barriers. Table 5 summarizes the main physiological barriers to oral drug delivery.

### 4.2. Advantages of Nanotechnology Application

By using nanotechnology and NPs, in addition to improving the solubility of bioactive compounds, their delivery, and their cellular uptake, it is possible to protect them from early degradation and rapid metabolism. Ellagic acid (EA), a polyphenol found in fruits and vegetables, whose several health-promoting properties are unfortunately associated with very poor solubility and many pharmacokinetic drawbacks, represents a typical example of a PHY engineered by nanotechnology. Several studies have reported on the development of appropriate nanomaterial-based devices that have been used to enhance EA solubility, its hydrophilic–lipophilic balance (HLB), and its GIT absorbability and/or to protect it from early metabolism [87,88,89,90,91,92,93]. In particular, EA with high water solubility was obtained using cyclodextrins [90,92], pectin [93], and polyester-based dendrimers [93].

Furthermore, by formulating PHYs using NPs, their controlled and targeted release can be realized, which is essential for effective administration. Controlled delivery results in a higher concentration at the target, thus allowing a reduction in the overall administered dose and consequently systemic toxicity [1]. Both internal and external factors, such as pH, temperature, ultrasound or magnetic field application, light incidence, and the type and physicochemical features of NPs, as well as the chemical structure and the physicochemical features of the bioactive compounds themselves, can control the specific release of bioactive compounds [1].

Stimuli-sensitive nano-capsules containing a bioactive derivative of paclitaxel and possessing an oil core showed the capability to improve the anticancer effects of the encapsulated compound taken by oral administration, thanks to targeted delivery and controlled long-term release [94]. The improved effects allowed a decrease in the dosage and the administration frequency, thus improving patient compliance [94]. The layer-by-layer self-assembly of pH-sensitive building blocks proved to be a promising approach to obtaining biomaterials with customized properties, which were successfully applied as stimuli-responsive nanocarriers [15]. Starting from biocompatible pH-dependent polyelectrolytes, nontoxic nanocarriers with high permeability were designed [94].

Additionally, the encapsulation of bioactive compounds in properly functionalized NPs can allow increased cellular uptake and slower drug release, thus improving the drug’s bioactivity and contributing to sustained therapy [17]. The effects of phospholipid composition on the pharmacokinetics and biodistribution of epirubicin-loaded liposomes were examined, proving a significantly prolonged circulating time, reduced clearance, and reduced heart toxicity [95]. Furthermore, carrying bioactive compounds in NPs can favor their distribution in specific brain areas, thus providing more valuable benefits in neuro-regenerative treatments while minimizing their accumulation in the systemic circulation and related toxic side effects [96]. Collectively, nanotechnology provides nanoformulation techniques, which, by using NS and NE approaches and/or different types of nanomaterials, enhance the solubility of bioactive compounds, PHYs included (Figure 7).

Due to the incessantly increasing interest in nanotechnology for manipulating poorly soluble bioactive compounds, including PHYs, and engineering them in more soluble and bioavailable dosage forms potentially suitable for clinical applications, the number of studies in the field has enormously improved in the last decade and especially in the past five years (Figure 2). In this regard, we thought that a single paper would not be sufficient to review all types of nanoformulation approaches recently developed to formulate PHYs. On the other hand, according to a recent research paper, up to the year 2019, while liposomes were the most studied NPs for nanoformulating PHYs, NEs were a less considered nanotechnological approach, and NSs were not even reported [97]. Additionally, since for their production, generally regarded as safe (GRAS) ingredients are usually employed, in addition to liposomes, NSs and NEs can be considered less toxic and the most suitable tactics for clinical application purposes. Accordingly, with the aim of emphasizing the potential of these promising nanoformulation techniques and stimulating scientists to further study and use them, we decided to focus the present review specifically on NS- and NE-based techniques.

### 4.3. Nanosuspension and Nanoemulsion Approaches

#### 4.3.1. Nanosuspension Techniques

These techniques are suitable for improving the solubility and bioavailability of both hydrophilic and lipophilic bioactive compounds. A nanosuspension consists of an aqueous colloidal dispersion of NPs stabilized by surfactants, co-surfactants, and polymers [17].

Drug-loaded NPs fabricated using this technique are able to overcome physiological barriers to oral delivery because they possess high dispersibility and solubility and can allow the sustained, controlled, and targeted delivery of the loaded drug, in addition to being endowed with improved stability and therapeutic effects [98]. NS techniques encompass both conventional and combined approaches. The conventional approaches to prepare NSs consist of bottom-up (B-U) and top-down (T-D) methods, as reported in Figure 8 and Table 6.

In particular, T-D techniques are referred to as physical methods, which start from particles with large dimensions and reduce their size to nanoscale dimensions via the media milling technique, HPH, UHPH, or supercritical fluid processes. In contrast, B-U methods are referred to as chemical and physical methods, which start by subjecting the atomic bioactive compound to precipitation, melt emulsification, coacervation, inclusion complexation, supercritical fluid extraction, or liquid antisolvent precipitation, causing self-association and self-organization to form nanosized materials [17,98]. In contrast, combined approaches mix a B-U phase, such as precipitation, with a subsequent T-D approach, such as HPH. They consist of the Nanoedge™ Technique (Baxter Healthcare), H 69 Technology (SmartCrystal^®^ technology group), H 42 Technology (SmartCrystal^®^ technology), H 96 Technology (SmartCrystal^®^, Abbott/Soliqs, Ludwigshafen, Germany), and Combination Technology (CTNO), as reported in Table 6.

In particular, the Nanoedge™ Technique combines the microprecipitation of the PHY in water and homogenization techniques, giving a better particle size distribution and better stability. Usually, precipitation is performed using water-miscible solvents, including methanol, ethanol, and isopropanol, and leads to the obtainment of an amorphous precipitate [17]. In the Nanoedge™ technology, an evaporation step is included to yield the solvent-free starting material, which is further processed by HPH using piston-gap homogenizers or by sonication. The homogenization phase allows the production of nanosized particles (80–700 nm) endowed with great stability in a short time and impedes further crystal growth [17]. H 69 Technology forms part of the SmartCrystal^®^ technology group and is like the Nanoedge™ approach, except for the immediate treatment of the micro-precipitate with cavitation, particle collision, and shear forces. Highly stable drug nanocrystals in the range of 20–900 nm can be obtained [17].

H 42 Technology belongs to the platform of SmartCrystal^®^ technology as well. In this case, the B-U step, which consists of PHY precipitation by SD carried out in aqueous media containing surfactants, is followed by the usual HPH phase (T-D phase) for further particle size reduction. Solvent-free dry intermediates and small drug nanocrystals are obtainable after a reduced number of HPH cycles (170–600 nm) and short processing times. Unfortunately, since high temperatures are necessary during SD, this technique is unsuitable for processing thermolabile compounds [17].

In contrast, H 96 Technology (SmartCrystal^®^, Abbott/Soliqs, Ludwigshafen, Germany) involves a B-U pre-treatment step by FD, followed by the usual T-D step for particle size reduction through HPH. In this case, H 96 technology is suitable for processing thermolabile or expensive drugs due to the low temperatures and the high yields of the FD [17]. Finally, combination technology (CTNO), without using organic solvents, combines two T-D approaches. In particular, a low-energy pearl milling phase of an aqueous macro-suspension that provides particles with sizes of 600–1500 nm is combined with the usual HPH phase, thus obtaining particles with a size of 250–600 nm. This approach permits a limited risk of crystal growth during storage, providing NPs with enhanced physical stability. In addition, to reduce processing times and costs, CTNO is suitable for scaling up [99].

#### 4.3.2. Manufacturing Methods

##### Antisolvent Precipitation Techniques

Precipitation is an economical and easily up-scalable B-U method, by which materials of sub-colloidal dimensions are induced to aggregate into colloidal-size particles. Briefly, supersaturated solutions of drugs in organic solvents miscible with water are slowly dispersed in water, acting as a non-solvent, with rapid stirring and at an optimized temperature [100]. Contact with the non-solvent causes supersaturation conditions, which provoke rapid nucleation and the slow growth rate of crystals. Thermodynamically stable crystals can be achieved, according to the classical Ostwald law of nucleation [101]. Additionally, crystal growth must be suppressed by the appropriate concentration of surfactant or selective crystallization inhibitors. Nanomorph™ (Soligs/Abbott/Patent No. D 19637517) is a marketed technology (Lucarotin^®^ (BASF, Ludwigshafen, Germany) based on this precipitation method and is used to produce amorphous drug nanoparticles.

##### Homogenization

Homogenization, generally referred to as a physico-mechanical process that can convert two immiscible liquids into a relatively homogenous mix, is a T-D technique used for particle size reduction. The mixing, blending, and stirring (agitation) of liquids, solids, as well as gases, are closely related to homogenization. Homogenization is an easily up-scalable technique and is adaptable to both dilute and concentrated suspensions. It is endowed with a low risk of contamination, thus making aseptic manufacturing possible [102]. In particular, by driving dispersion, suspension, or emulsion at high velocity through a narrow passage, acting as a disruption valve, microparticles (<25 µm) are broken down to particles of nanosized dimensions with a narrow particle size distribution [102]. Homogenization methods include conventional homogenization (CH), usually up to 50 MPa, and (ultra-) high-pressure homogenization ((U)HPH), which differs from CH in the maximum pressure level reached and in being dependent on the homogenizer design and characteristics, such as gap size, seals, and valve geometry [103]. In particular, UHPH reaches pressure levels up to 400 MPa, while HPH reaches pressure levels between 50 and 200 MPa [103]. During homogenization, shear forces help to fracture particles with crystal defects, while the use of viscosity enhancers can inhibit crystal growth, thus improving the process of nanosizing [104]. Homogenization can improve metastable amorphous particles prepared by the precipitation method previously described to stable crystal forms. Generally, multiple cycles and high energy consumption are required to produce particles in the desired size ranges. Hydrosol (Novartis/Patent No. GB 2269536), Nanocrystal™ (Elan Nanosystems/Patent No. US 5145684), Dissocubes^®^ (Skye Pharma/Patent No. US 5858410), Nanopure (Pharma Sol/Patent Application No. PCT/EP00/0635), and NANOEDGE™ (Baxter/Patent No. US 6884436) are among the patented or patent-pending homogenization technologies.

##### Wet or Media Milling

Wet or media milling is a T-D method that provides NSs by using a high-shear ball mill or media mill. Briefly, a mixture of drugs, stabilizer(s), and water loaded within the mill chamber is subjected to both impacts and attrition, which cause particle reduction. The method requires low energy consumption, is easily up-scalable, allows minimum batch-to-batch variation, and permits handling large quantities of material. Unfortunately, the wear and tear of milling media may occasionally introduce residues in the finished product. Such a drawback can be minimized by using a highly crosslinked polystyrene resin milling medium [105].

##### Dry Co-Grinding Method

Dry co-grinding T-D methods using proper polymers and copolymers are able to provide stable NSs in a more economical way than wet grinding processes without the addition of any toxic solvents [106]. This method allows the improvement of surface polarity and the modification of the crystalline state of the original drug to amorphous forms, thus improving the saturation solubility and hence the dissolution rate of poorly soluble drug nanosuspensions [107].

##### Liquid Emulsion Evaporation Technique

In this B-U technique, an emulsion is formed by dissolving the drug in an organic solvent or cosolvents, thus obtaining a solution that is dispersed in an aqueous phase containing a proper surfactant. Upon rapid evaporation of the solvent under reduced pressure, the desired NS can be obtained.

##### Sonoprecipitation Method

Sonoprecipitation is a novel B-U approach allowing the fabrication of stable NSs by employing ultrasound in the frequency range of 20–100 KHz, which enables particle size reduction and the control of the size distribution of the active pharmaceutical ingredient. The technique is effective for minimizing the nucleation and crystallization processes [108].

#### 4.3.3. Nanosuspension-Based Phytochemical Delivery Systems

Table 7 reports some examples of PHYs nanoformulated using both conventional and combined nanosuspension techniques.

Considering the most recent case studies reported in Table 7, the poorly water-soluble flavonoid extracted from licorice root, namely, isoliquiritigenin, effective against several forms of cancer, was nanoformulated by Qiao and colleagues using a T-D approach [115]. Hydroxypropyl cellulose-SSL (HPC-SSL) and polyinylpyrrolidone-K30 (PVP-K30) were used as stabilizers, and particles with mean sizes of 238 nm and 354 nm, respectively, were obtained. Both NSs showed a lamelliform or ellipse shape and a higher dissolution rate of isoliquiritigenin, thus demonstrating enhanced cellular uptake and improved cytotoxicity [115]. Additionally, while the developed NSs caused an apoptosis rate 7.5–10-fold higher than that caused by the non-formulated isoliquiritigenin, toxicity in normal human embryonic lung fibroblasts (HELFs) was lower [115].

Furthermore, celastrol (CSL), which is one of the main components of *Tripterygium wilfordii* Hook f., having significant antitumor activity but poor solubility, low oral bioavailability, and systemic toxicity, was nanoformulated by Huang et al. by way of a B-U technique [116]. In particular, through an antisolvent precipitation method with poloxamer 188 (P-188) as a stabilizer, CSL nanosuspensions (CSL-NSs) containing nanosized spherical-shaped particles were prepared with high EE (98%) and DL (87%). Upon its nanoformulation, CSL dissolution in vitro was greatly enhanced, and its cumulative drug release reached approximately 69.20% within 48 h [116]. Additionally, in in vivo experiments, CSL-NSs (3 mg/kg, i.g.) displayed a significantly enhanced tumor inhibition rate (TIR) in comparison with that of the CSL suspension when administered orally [116].

SC-CO_2_ extracts obtained from the de-oiled *C. longa* Linn (turmeric) rhizome were converted to NPs by using a B-U nanosuspension technique, performing a supercritical fluid expansion method using SC-CO_2_ [117]. The production of particles was based on the expansion of the supercritical solution and provided nanosized, almost spherical particles with significantly improved dissolubility [117].

The poor aqueous solubility and low oral bioavailability of naringenin (NRG) were addressed by preparing NRG nanosuspensions (NRG-NS) using polyvinylpyrrolidone (PVP K-90) and Tween 80 as stabilizers via an antisolvent sonoprecipitation method [118]. Optimized conditions provided NRG-NSs with the smallest particle size of 117 nm and a zeta potential of −15 mV in an amorphous form possessing higher absorption in the GIT, as well as an improved dissolution rate and oral bioavailability [118].

More recently, NRG was nano-suspended by applying a precipitation–ultrasonication method using different surfactants and polymers, such as sodium cholate (SC), sodium lauryl sulfate (SLS), polyethylene glycol 4000 (PEG), polysorbate 80 (Tween^®^ 80), poloxomer-188, and D-α-tocopherol polyethylene glycol 1000 succinate (TPGS or Vitamin E-TPGS) [119].

The best nanoformulation showed small particles with a size of 118 nm, an increased drug dissolution rate in simulated gastric fluid at pH 1.2 (SGF) and phosphate buffer at pH 6.8 (PB), and an improved pharmacokinetic profile compared to pure NRG and was stable over a period of six months [119].

As a continuation of their previous research, Rajamani et al. prepared NRG-loaded NSs using TPGS to evaluate the ability of the TPGS-coated NRG-NSs to reverse the drug resistance in human breast adenocarcinoma MCF-7 cells and animal models [120]. Treatment with NRG-NSs significantly increased intracellular ROS levels, the mitochondrial membrane potential, caspase-3 activity, lipid peroxidation status (TBARS), and decreased GSH levels when compared to free-NRG treatment in MCF-7 cells while exhibiting dose-dependent in vitro antitumor activity on DLA cells [120]. Additionally, a significant increase in life span, associated with a decrease in the cancer cell number and tumor weight, was noted in mice [120].

By using a precipitation-combined ultrasonication method, glaucocalyxin A (GLA), which is a PHY component with multiple pharmacological activities affected by poor solubility, was formulated in NSs [121]. GLA-NSs were obtained as spherical particles with a smooth surface, small size (143 nm), and a DL% of about 9%. In contrast to the free-drug solution, GLA-NSs showed higher in vitro antitumor activity against HepG2 cells (IC_50_ value of 1.793 vs. 2.884 µg/mL at 24 h, *p* < 0.01) and better anticancer efficacy in H22-bearing mice (54.11% vs. 36.02% tumor inhibition rate) [121].

Furthermore, sucrose ester (SE)-stabilized oleanolic acid (OA) NSs (SEOA NSs) for enhanced delivery were prepared via organic solvent evaporation methods, producing spherical SEOA NS particles (~100 nm in diameter) that were stable for a month at 4 °C. The best-performing SEOA 4121 NSs showed a great increase in saturation solubility (1.89 mg/mL vs. 3.43 µg/mL), the dissolution rate, cytotoxicity, bio-efficacy, and bioavailability [122].

NS-based formulations of the flavonoid-rich fraction of *P. guajava* L. extracts with enhanced antihyperglycemic activity and the best physical parameters were prepared using PVA through the nanoprecipitation method and tested in vivo on high-fat-diet-fed, streptozotocin-induced type 2 diabetic animals [123]. Upon oral administration, the developed NSs restored the normal level of blood glucose in the first hour and showed beneficial effects on various hepatic and renal parameters [123]. Additionally, NSs enhanced the absorption, decreased the metabolism, and improved the stability of flavonoids [123].

*Nigella sativa* L.-based NSs were prepared, and their composition, as well as their bioactivities in terms of antioxidant, antidiabetic, antibacterial, and hemolytic activities, was investigated and compared with those of the non-formulated ethanolic extract [124]. The results revealed that the NSs of *N. sativa* seeds showed higher total phenolic and total flavonoid contents than those of the ethanolic seed extract. NSs showed higher antioxidant and antidiabetic activities, as well as biofilm inhibition activity against *Escherichia coli*, than those of both the extract and ciprofloxacin. Additionally, the study showed that NSs enhanced the bioavailability of bioactive plant compounds as compared to the ethanolic extract [124].

#### 4.3.4. Emulsion-Based Techniques

As suspension methods, emulsion techniques can be used to reduce the particle size of both hydrophilic and hydrophobic bioactive compounds, thus achieving orally administrable formulations, which are very promising for pharmaceutical applications. NEs can more easily bypass the physiological barriers to oral delivery due to their improved solubility and bioavailability [1,125]. Emulsion technology involves the encapsulation of bioactive compounds in small droplets, mixing an aqueous phase (w) with an oil phase (o) and obtaining either water-in-oil (w/o), oil-in-water (o/w), or bi-continuous colloidal dispersions, which are stabilized using specific additives, such as GRAS pharmaceutical surfactants, co-surfactants, and emulsifiers (Table 8) [1,126]. In particular, NEs are multifunctional lipid nanosystems that integrate enhanced permeability, tissue and cell targeting, imaging, and therapeutic functions. By strategically selecting edible oils, surfactants, and surface modifiers and choosing different types of payloads, as well as implementing a rationale design, multifunctional NEs can work as a safe and effective delivery system that can cross oral barriers [79].

Emulsions encompass microemulsions (micro-sized droplets, not considered in this paper), NEs (100–500 nm droplets), and self-emulsifying drug delivery systems (SEDDSs), which in turn include self-nanoemulsifying drug delivery systems (SNEDDSs) and self-microemulsifying drug delivery systems (SMEDDSs), so classified based on the dimensions of their NPs. Additionally, self-double-emulsifying drug delivery systems (SDEDDSs) represent a further evolution of conventional SEDDSs (Table 8) [1].

By using NE techniques, it is possible to obtain PHY-based formulations characterized by particles with sizes of 100–500 nm endowed with improved solubility, stability, and bioavailability and an extended half-life. Upon the use of suitable additives (5–10%), isotropic, transparent, and kinetically stable suspensions are achievable [127,128]. NEs are generally prepared using either low-energy techniques (LETs), not involving mechanical devices, or high-energy techniques (HET), requiring the use of mechanical devices and strong agitation [126].

Among NEs, SEDDSs are anhydrous nanodispersions obtained by spray drying or freeze drying a mixture of an oil phase, surfactants, co-surfactants/cosolvents, and a lipophilic bioactive compound. SEDDSs are particularly suitable for orally delivering lipophilic bioactive compounds because they spontaneously arrange in colloidal emulsions when mixed with water or with fluids in the GIT after the ingestion of capsules filled with SEDDSs [1,129]. SEDDSs include SNEDDSs (droplets size < 50 nm), SMEDDSs (droplets size of 100–200 nm), and SDEDDSs. The latter can form water-in-oil-in-water (w/o/w) or oil-water-in-oil double emulsions in GIT fluids, thus representing novel self-emulsifying formulations, expressive of the further evolution of conventional SEDDSs [130].

##### High-Energy and Low-Energy Methods

High-energy methods involve the use of a mechanical device, such as high-pressure valve homogenizers, microfluidizers, and ultra-sonicators, while low-energy methods use the energy input deriving from the chemical potential of the components used to form NEs. In the latter case, the NEs form at the oil-and-water-phase interface with the gentle mixing of the components, and its formation depends on factors such as temperature, composition, and compounds’ solubility. Table 9 reports the main adopted high-energy and low-energy methods and their recent applications in nanoformulating PHYs.

In the following sections, except for HPH, which has been described in Section 4.3.2, the methods reported in Table 9 are discussed.

##### Microfluidization (MF)

Microfluidization (MF) is a non-thermal HPH capable of providing fine emulsions by processing a fluid at high shear using a microfluidizer. The technique is commonly applied in the food, cosmetic, and pharmaceutical industries. In particular, in the chamber of the microfluidizer, mechanical energy is transferred to the product under very high pressure (up to 200 MPa), thus creating very fine and stable emulsions with reduced particle size and an increased surface area [141]. To intensify the exposure time and homogenization effect, suspensions can be passed through the microfluidizer chamber multiple times. MF has solved typical issues of emulsion instability, including creaming, sedimentation, flocculence, Ostwald ripening, and coalescence [142]. A microfluidizer, such as Microfluidizer M-110P (Microfluidics^®^ high-shear fluid processors, Microfluidics, Westwood, MA (USA), is composed of a hydraulic power system, which provides power to the microfluidizer equipment; a high-pressure single-acting intensifier pump, which multiplies the pressure supplied from the hydraulic power system; a pneumatic pump, which pressurizes the air up to about 276 MPa; and an interaction chamber [143]. In particular, the interaction chamber, available in two different types (Y-type and Z-type), is a continuous micro-channel responsible for turbulent mixing, energy dissipation, and generating a homogeneous pressure profile, thus helping in obtaining particles with a narrow size distribution. An optional auxiliary module can cool the product stream back to room temperature while it exits the system. The fluid is forced into the interaction chamber at very high pressure with high velocity, where it is divided into two streams that collide at 180° in the interaction chamber. Due to a rapid drop in pressure, impact, cavitation, shear, and turbulence effects are observed, and emulsification occurs [143].

##### Ultrasonication (US)

Ultrasonication is one of the most efficient homogenizations HETs currently used to produce NEs, with the potential for aseptic processing. Ultrasonic homogenizers for commercial applications are able to produce small droplets at relatively low operating costs with low energy consumption and are easily cleaned [144]. The cavitation forces produced by high-intensity ultrasonic waves lead to the formation of microbubbles in the liquids that oscillate rapidly and eventually implode, thereby generating intense disruptive forces that lead to the formation of small emulsion droplets. Ultrasonic homogenizers can be used to produce NEs on different scales [144].

##### Phase Inversion Composition (PIC) and Phase Inversion Temperature (PIT)

PIC and PIT are phase inversion emulsification techniques characterized by low energy requirements [145]. In both processes, inversion between the continuous and dispersed phases is achieved by either changing the composition (PIC) or temperature (PIT), thus obtaining finely dispersed emulsions in a more sustainable way compared to classical HPH methods [145]. In PIC, emulsion phase inversion is reached by changing the water/oil ratio, while in PIT, phase inversion is brought about by changing the surfactant affinity for the two phases upon heating [145].

##### Spontaneous Emulsification (SE)

Spontaneous emulsification is a LET that is extensively used in several applications [146]. In SE, droplet formation can be achieved by mixing two immiscible liquids that are not in equilibrium without the need for external energy input. In fact, if the immiscible liquids are in thermodynamic equilibrium, the large and positive energy to expand the interface cannot be compensated by the small and positive entropy of dispersion, thus requiring external energy input to achieve emulsification [147]. In contrast, if the immiscible liquids are not in equilibrium, gradients in chemical potential between the phases may produce negative values of the free energy of emulsification, and emulsification occurs spontaneously [147].

##### Novel Nanoemulsion Preparation Techniques

New emulsification approaches are being increasingly developed to increase the range of materials that can be formulated and the range of available operating conditions and to simultaneously lower production costs. Water-in-oil NEs can be prepared by condensing water vapor on a subcooled oil-surfactant solution. The NEs formed using the condensation approach have dimensions of about 100 nm. In particular, an oil bath is placed in a humid environment with an appropriate concentration of a surfactant, and upon decreasing the temperature below the dew point, water condensation is induced on the oil surface, resulting in NE formation. The process is simple, rapid, scalable, and energy-efficient, with potential applications for processed foods [148]. Additionally, Pickering nanoemulsions (PNEs) are NEs stabilized by solid particles (for example, colloidal silica), which adsorb onto the interface between the water and oil phases [149]. Typically, the emulsions are either water-in-oil or oil-in-water emulsions, but other more complex systems, such as water-in-water, oil-in-oil, water-in-oil-in-water, and oil-in-water-in-oil, also exist. The use of PNEs overcomes the problems associated with surfactant desorption and Ostwald ripening. The low-energy approach cannot be used to produce PNEs, while the high-energy approach prevents the adsorption of the particles on the droplets. The vapor condensation method used for PNE preparation is a single-step process and has several advantages over conventional techniques, such as the use of low concentrations of NPs [150].

#### 4.3.5. NE-Based Phytochemical Delivery Systems

NE-based DSs have been exploited for formulating herbal drugs, whole plant extracts, a single PHY, or PHY mixtures, which are poorly insoluble, unstable in highly acidic pH, and metabolized by the liver if orally administered in the free form. Interestingly, NE-based DDSs can minimize side effects due to possible drug accumulation in non-targeted areas, so their oral administration is also authorized in pediatric and geriatric individuals [151]. For example, NE techniques were used to nanoformulate turmeric, curcumin (diferuloylmethane), and di-benzoyl-methane (a structural analog of curcumin). Tannins, stilbenes, and flavonoids, which demonstrated in vitro antioxidant effects, have been encapsulated in nano-drops by using NE methods [1]. In addition, bioactive lipids and carotenoids were formulated as NEs, with, respectively, higher stability against autoxidation and increased bio accessibility observed [1].

In this regard, curcumin-loaded lipid NEs (CmLN) functionalized with a nona-arginine peptide (R9-CmLN) were prepared by Simion and colleagues, using triacylglycerol as the oil phase and Tween 20 as the emulsifier [151]. When used at a therapeutically relevant concentration, R9-CmLN demonstrated low hemolytic activity, low cytotoxicity, and anti-inflammatory effects. Additionally, in vivo biodistribution studies in mice revealed a high accumulation of R9-CmLN in the liver and the lungs, suggesting its potential therapeutic applications in different inflammatory pathologies localized in such organs [151].

The limited efficacy of curcumin due to its low oral bioavailability was overcome by the group of Nazari-Vanani by developing a SNEDDS [152]. An optimal formula for the SNEDDS comprised ethyl oleate/Tween 80/PEG 600 (50/40/10 *w*/*w*), which formed 11.2 nm uniform droplets when subjected to mild agitation. In in vivo experiments in rats orally administered the SNEDDS, the curcumin C max was increased by 3.95 times, while its bioavailability was enhanced by 194.2%, compared to the curcumin suspension in water [152]. In another study, to enhance the bioavailability of curcumin and its impact on the levels of docosahexaenoic acid (DHA), which is an important long-chain omega-3 polyunsaturated fatty acid (PUFA), Sugasini and Lokesh developed a curcumin-loaded NE using a phospholipid core material (Lipoid™) [153]. In particular, curcumin was dissolved in coconut oil, sunflower oil, or linseed oil, and the NEs were obtained after mixing with Lipoid™ using HPH. Experiments in rats demonstrated high levels of curcumin in the serum, liver, heart, and brain and a significant increase in DHA levels in serum and lipid tissue [153].

Moreover, curcumin-loaded NPs were prepared by using a particular emulsion technique referred to as an emulsion–diffusion–evaporation method [154]. Briefly, curcumin was dissolved in acetone and ultrasonicated, stirred for 1 h at 55 °C, and finally heated in an oven until the complete evaporation of the organic solvent, resulting in 32 nm sized NPs [154].

Experiments carried out on diabetic rats evidenced a significant reduction in the blood glucose level while increasing that of insulin in the group treated with the developed curcumin-enriched NPs [154].

Aiming at developing an effective anticancer agent against oral squamous cell carcinoma (OSCC), curcumin was formulated as curcumin-loaded lipid NEs (CUR-NEs), obtaining 100 nm sized particles. In in vitro investigations on OSCC HSC-3 cells, CUR-NEs exhibited significant cytotoxic effects on OSCC cells in a dose-dependent manner compared with the control [155].

Hu and co-authors manufactured an SDEDDS loaded with both epigallocatechin-3-gallate (EGCG) and α-lipoic acid (ALA) (EA-SDEDDS), having improved photo-stability relative to free EGCG and equal antioxidant activity to that of a solution of EGCG and ALA [156]. In particular, a modified two-step method was used and optimized to prepare EA-SDEDDS. In the first step, primary emulsification was achieved by adding the aqueous phase containing EGCG to the oily phase consisting of macadamia oil, cetostearyl alcohol, ALA (6 g/L), and polyglycerol polyricinoleate (PGPR) as a hydrophobic emulsifier using an overhead stirrer. In the second step, the primary emulsion was further mixed with different types of hydrophilic emulsifiers (S721, P10, L23, and S40) [156].

In a further study, the poor in vivo antioxidant activity of EGCG was significantly increased when it was formulated as an NE by Koutelidakis et al. [157]. In particular, in a typical experiment, w/o, o/w, and double emulsions were prepared and administered to mice by gavage. Two hours after administration, the total antioxidant capacity (TAC) was measured with ferric-reducing antioxidant power (FRAP) and oxygen radical absorbance capacity (ORAC) assays in the plasma and some tissues (especially the colon, jejunum, heart, and spleen). While no toxic effects were observed, EGCG emulsion II (o/w), which contained 10% olive oil and 0.23 mg/mL esterified EGCG in the fatty phase, exerted an antioxidant effect in mouse plasma that was remarkably higher than that of the aqueous solution of EGCG. Additionally, in several tissues of mice administered emulsion II, values of TAC higher than those observed in animals treated with emulsions I and III were observed [157].

Table 10 reports some examples of plant-derived bioactive compounds nanoformulated by using NE techniques to obtain emulsion-based DSs.

Naringenin and hesperetin are citrus flavonoids possessing well-documented protective effects on the cardiovascular system. Unfortunately, their poor water solubility, affecting their bioavailability, strongly restricts their therapeutic use. To address these issues, they were recently encapsulated into lipid NEs (LNEs), resulting in flavonoid-loaded LNEs that had nanosized particles with sizes of 190–200 nm (naringenin) and 193–218 nm (hesperetin), negative zeta potential, an EE of over 80%, good in vitro stability, and steady release of the cargo. Additionally, while the LNEs did not exhibit in vitro cytotoxicity and did not provoke the lysis of mouse erythrocytes, they exerted significant anti-inflammatory effects [165].

The potential of NE techniques as enhancers of drug solubility, oral bioavailability, and stability was demonstrated by Yin et al. when they prepared NE-based baicalein DSs [166]. In particular, baicalein (BCL), possessing important pharmacological activities but poor solubility and low stability in the GIT, was formulated using an NE technique, in which an HPH process was exploited to minimize the quantity of surfactants [166]. BCL-loaded NEs were obtained, which demonstrated ~90 nm sized particles, EE > 99%, and an oral bioavailability of BCL 525% higher than that of BCL suspensions. Additionally, BCL-loaded NEs exhibited excellent intestinal permeability and transcellular transport ability, while cytotoxicity was acceptably low for oral purposes [166].

Liang and co-workers used NE techniques to formulate imperatorin, having antitumor, antibacterial, anti-inflammatory, anticoagulant activities, and myocardial-hypertrophy-inhibitory effects [167]. An optimized preparation required 1.39 g of egg lecithin, 0.21 g of poloxamer 188, and 10.6% soybean oil as stabilizers and the oily phase, respectively, thus providing imperatorin-loaded spheres, showing round globules with a relatively uniform shape and sizes within 200 nm. The imperatorin-loaded lipid NPs allowed significantly enhanced bioavailability of imperatorin and inhibited MDA-MB-231 cell proliferation, thus making them promising for the treatment of late-stage breast cancer [167].

The PHY constituents of *Pandanus conoideus* Lamk (red fruit) are endowed with significant antitumor activity against breast cancer but are poorly adsorbable in the GIT. To address this issue, Satria and co-workers prepared SNEDDS-type formulations (particle size 193 nm) of the *Pandanus conoideus* Lamk (red fruit)’s red oil extract [168]. Once tested in vitro against MCF-7 breast cancer cell lines, the red-oil-loaded SNEDDS formulations showed good cytotoxic activity, higher than that demonstrated by the non-formulated *P. conoideus* extract at the equivalent dose [168].

More recently, the red oil (*P. conoideus*) previously reported was formulated by using the NE technique in the form of a conventional NE, a cream NE, and an NE gel, intended for skin application. The NEs were prepared by employing sucrose palmitate as an emulsifying agent and a brute force method using an Ultra-Turrax homogenizer as a high-speed mixer [169].

The red fruit oil-based conventional NE showed pseudoplastic flow properties, a spherical shape, and an average particle size of 103 nm. The cream NE demonstrated plastic flow properties, and an average particle size of 392 nm, while that in the form of gel revealed plastic flow properties and an average particle size of 144 nm. In antioxidant experiments using the 2.2-diphenyl-1-picrylhydrazyl (DPPH) assay, the cream and gel nanosuspensions showed IC50 values of 6.14 and 48.85, respectively [169].

Likewise, herbal drugs such as *Plantago lanceolate* [170], ethyl acetate extracts of bay leaves (*Eugenia polyantha* Wight) [171], myricetin [172,173], quercetin [174], and baicalin [175] were also developed in SNEDDS formulations (Table 10) to increase their solubility, permeability, bioavailability, and pharmacological effects.

In particular, the low oral bioavailability of myricitrin was improved by preparing myricitrin (M)-loaded SMEDDS consisting of an oil phase (ethyl oleate), a surfactant (Cremophor EL35), and a co-surfactant (dimethyl carbinol) [173]. The prepared M-SMEDDS exhibited stable physicochemical properties, small droplets (22 nm), a negative zeta potential (−23 mV), and high EE (92.7%) [173]. The in vitro release study showed that the release of myricitrin from M-SMEDDS was significantly higher than that from a free myricitrin solution, while the oral bioavailability of M-SMEDDS was 2.47-fold higher than that of the free drug [173].

Some natural isothiocyanates (ITCs), including sulforaphane (SFN) (isolated from broccoli in 1992), allyl isothiocyanate (AITC) (abundant in mustard, horseradish, and wasabi), benzyl isothiocyanate (BITC) from garden cress, and phenethyl isothiocyanate (PEITC) from watercress, demonstrated a plethora of pharmacological effects [176]. Currently, some of them are in phase I and II clinical trials to assess their safety, tolerance, pharmacokinetics, and therapeutic benefit in the context of different types of cancer, diabetes, kidney disease, skin disorder, blood/vascular disease, asthma, and autism [176].

Unfortunately, the results of these clinical trials only partially confirmed the promising beneficial potential demonstrated during in vitro studies due to their instability, their low bioavailability, and, concerning cancer, the influence of the complex tumor microenvironment [176].

In the last five years, Encinas-Basurto et al. encapsulated AITC in poly (lactic-glycolic acid) nanoparticles (PLGA NPs) to extend its shelf life and enhance its antiproliferative properties using an emulsion solvent evaporation method [177]. The obtained AITC PLGA NPs had a particle size of about 200 nm, polydispersity >2%, and negative zeta-potentials (−8.0 mV). These NPs demonstrated reduced degradation, decreased volatility, and an extended shelf life when compared with free AITC [177]. In vitro experiments on cancerous HeLa and MDA-MB-231 cells showed that the sustained release of AITC from polymeric NPs resulted in significant toxicity toward tumor cells. Subsequently, the same group modified the surface of AITC-loaded PLGA NPs using a specific antibody to target the epidermal growth factor (EGF) receptor (EGFR) overexpressed on epithelial squamous carcinoma cells [178]. AITC-loaded PLGA NPs showed more effective anticancer properties when compared with free AITC. The attachment of the anti-EGFR antibody to the NPs’ surfaces further enhanced their cytotoxicity toward the tumor cells and reduced toxicity against normal cells [178].

Kumar et al. encapsulated BITC in an NE through ultrasonication using Tween 80 or decyl-β-d-glucopyranoside as stabilizers [179]. The average size of NE particles was about 32 nm, and EE was 99%. The nano-DDS showed good stability at pH 5, 7, and 9, while in highly acidic or basic conditions (pH 2 and 12), aggregation occurred, probably due to ineffective electrostatic repulsion and hydrolysis [176].

In recent research, Uppal et al. developed cerium oxide NP (CONP)-based DDSs using the ultrasonic nanoemulsification method [180]. The synthesized NPs (size ≤ 5 nm) were then loaded with BITC. The average particle size of BITC-loaded CONPs was about 5 nm, while the zeta-potential value was about −15 mV. The formulation obtained showed high EE and good DL, while it was demonstrated to significantly inhibit the viability of MDA-MB-231 cells. The same group also produced a new BITC-based NE system employing the heating stirring–sonication method and rhamnolipid as a biosurfactant [181]. The optimized NE exhibited good long-term stability, very high EE, the sustained release of BITC, and increased cytotoxicity against MDA-MB-231 cells as compared to BITC alone [181]. Collectively, from data in the literature, NE techniques are the main method used for the nanoformulation of BITC, providing BITC-loaded NPs with very high EE%, enhanced absorption and bioavailability, a prolonged shelf life, and the sustained release of BITC, as well as improved anticancer activity against several cancer cell lines [181].

Oil-in-water (o/w) NEs composed of *Satureja montana* essential oil (SEO), owing to the high content of active PHYs and several biological effects, were prepared using Tween 20 or Tween 80 as emulsifiers [182]. The achieved SEO-NEs were analyzed in terms of the hydrodynamic diameter, zeta potential, and polydispersity index, which confirmed the formation of stable NEs homogeneous in size. Microbiological experiments carried out on Gram-positive and Gram-negative clinical isolates established that the NE-based formulation preserved and improved the antimicrobial activity of pristine SEO [182].

More recently, Rinaldi and colleagues formulated SEO, preparing and optimizing o/w NEs composed of SEO and Tween 80 and obtaining 112 nm sized NPs [183].

Minimal inhibitory concentrations (MICs) and minimal bactericidal concentrations (MBCs) evaluated by the microdilution method showed that the SEO-based NEs exhibited higher inhibitory effects against planktonic *E. coli* than SEO alone. Additionally, SEO formulations enabled an efficient reduction in the biofilm produced by strong producer strains at sub-MIC concentrations. Given these results, SEO-based NEs could be promising to ensure food safety and quality and to counteract the antibiotic resistance of poultry-associated *E. coli* if applied/aerosolized in poultry farms [183].

A SNEDDS for RES capable of exerting anti-fatigue activity was developed by Yen et al. to improve RES bioavailability and was evaluated for its anti-fatigue activity in rats. The optimized SNEDDS was composed of Capryol 90, Cremophor EL, and Tween 20 and showed nanosized particles of approximately 41 nm. This RES-SNEDDS not only enhanced the oral bioavailability of RES upon administration in rats but also exerted an improved anti-fatigue pharmacological effect [184].

More recently, RES-loaded SMEDDSs with particles in the range of 22–26 nm have been developed with (SMEDDS-1) and without inhibitory excipients (SMEDDS-2) to increase RES’s oral bioavailability by inhibiting intestinal metabolism [185]. The results demonstrated that while similar physicochemical properties between inhibitory SMEDDS-1 and non-inhibitory SMEDDS-2 were observed, the bioavailability of RES was increased to 76.1% in SMEDDS-1 [185].

Astaxanthin and α-tocopherol, which showed wound-healing activity in diabetics, were recently formulated with κ-carrageenan to obtain bicomponent NEs (AS-TP@KCNEs) [186]. In vitro and in vivo experiments on diabetic mice demonstrated that AS-TP@KCNEs were biocompatible and possessed healing properties that accelerated wound closure and exhibited better control of hyperglycemia, thus reversing diabetes mellitus complications [186].

Spices have been known to exert numerous functions that are useful against different diseases, along with strong anticancer potential. In particular, clove and turmeric are spices with strong anticancer potential.

In this context, Nirmala et al. developed an oil-based NE of cloves (*Syzygium aromaticum*) buds and tested its anticancer efficacy against thyroid cancer cells (HTh-7). The clove-loaded NE showed antiproliferative effects against thyroid cancer cells, with apoptosis seen as the mode of cell death [187].

More recently, to improve the medicinal properties of *Syzygium aromaticum* L., *S. aromaticum* L. bud EO was nanoformulated using the ultrasonication NE technique to obtain nanosized DSs (SABE-NE) with 131 nm sized particles [188]. While the produced SABE-NE induced an apoptotic response and significant cell death in HT-29 cancer cells, normal HFF cells exerted limited cytotoxic impacts. Moreover, in vivo tests on mouse livers demonstrated the cytoprotective properties of SABE-NE [188].

Since kaempferol (KPF) has been reported to induce glioma cell death, Colombo et al. prepared NEs containing KPF with and without chitosan to investigate their potential for KPF brain delivery following intranasal administration and to evaluate their antitumor activity against glioma cells [189]. KPF-loaded NE (KPF-NE) and KPF-loaded mucoadhesive NE (KPF-MNE) were prepared by using the HPH technique, which demonstrated significantly higher permeation capability across the mucosa in ex vivo diffusion studies [189]. Both types of KPF-NE were safe for the nasal mucosa and able to preserve KPF’s antioxidant capability. Additionally, KPF-MNE significantly enhanced the amount of drug delivered into the rat brain following intranasal administration and reduced C6 glioma cell viability through the induction of apoptosis to a greater extent than either free KPF or KPF-NE [189].

Other case studies have been reported in the literature, as listed in Table 10, on the formulation of β-carotene, astaxanthin, curcumin, ginger EO, and capsaicin with NE technology using HPH, SE, US, or MF methods and obtaining nanosized droplets with significantly improved physicochemical and biological properties [190,191,192,193,194,195,196,197,198].

#### 4.3.6. NS- and NE-Based Phytochemical Formulations: The State of the Art in Graphs

In this section, we provide some graphical interpretations of information reported in the previous sections. In particular, Figure 9 presents the scenario concerning the main NS- and NE-based PHY-loaded formulations developed in recent years grouped according to their PHY contents.

The results evidence that NE-based technologies are the most widely adopted and studied (75.6%) relative to NS-based ones (24.4%). Secondly, concerning the types of plant-derived bioactive compounds used to prepare NE-based formulations finalized to treat human diseases, EOs and curcumin are the most frequently chosen (11.5 and 10.3%, respectively), followed by carotenoids (7.7%). In contrast, concerning the NS-based formulations considered here, naringenin (3.8%), quercetin (2.6%), and resveratrol (2.6%) are the most commonly used PHYs, while no NS-based formulations of EOs have recently been developed.

Figure 10 shows that B-U methods (63.2%) are preferred to develop NSs containing PHYs, followed by CTNIs (2.6%) and by T-D ones (only 1.1%).

Among the three main types of NEs reported, the most developed in the literature are the conventional ones (65.2%), and SNEDDSs (including SMEDDSs, SNEDDSs, and SDEDDSs) account for 32.6%, while MEs make up only 2.2% (Figure 11). Figure 12 shows the frequency with which HETs are employed to develop NEs vs. that of LET, thus evidencing that the first are preferred to the second ones (64.3 vs. 35.7%).

Finally, among HETs, US and SE are the most applied (28.2%), followed by HPH (25.6%), while the frequency of utilization of MF (10.3%), PIT (5.1%), and PIC (2.6%) is remarkably lower (Figure 13).

## 5. Nanomaterials and Nanoparticles: What We Know and What We Should Know

The use of nanotechnology in products regulated by the FDA, such as foods, cosmetics, medical devices, and drugs, has endured for several decades. According to the FDA’s Center for Drug Evaluation and Research (CDER), drug products containing nanomaterials are very different from conventional ones and should receive particular attention. Since the early 1970s, there has been a continual increase in the number of approved drug products containing nanomaterials, and more than 60 applications have been approved so far, but interest continues to rise [199].

Together with liposomes and nanocrystals, NEs are among the most common types of drug products containing nanomaterials being approved due to the use of GRAS material for their development. Drug products containing nanomaterials are unique and possess nonpareil chemical, physical, or biological properties differing from those possessed by traditional drugs [199]. Importantly, the presence of nanomaterials in a drug formulation may positively or negatively impact the quality, safety, or efficacy of the product, mainly because drug products containing nanomaterials may follow a different pathway in the body compared to that of a non-nanosized drug [199] (Table 11). After a drug product formulated as NPs enters the bloodstream, it can interact with specialized immune cells called macrophages, which can engulf and transport it to the target site, such as that where bacteria, fungi, viruses, or tumor cells reside. In contrast, these areas are typically difficult to reach for a non-nanoformulated drug. In addition, a drug formulated as NPs often has a special coating that can prevent it from an immune cell attack, thus having the ability to circulate in the bloodstream for a prolonged time and to reach untouched tumor tissues or infected areas. The ability of nanoformulated drugs to target areas of the body and to bypass others can significantly reduce the risk of side effects, such as toxicity to nontarget organs, and potentially increase the effectiveness of the treatment. For these reasons, nanomaterials are most frequently used to formulate drugs intended for the treatment of cancer or infections [199].

### 5.1. Ongoing Actions to Address Challenges Related to Nanotechnology

With the aim to inform agency guidance and regulatory review, the Office of Testing and Research (OTR) in CDER’s Office of Pharmaceutical Quality has been conducting research to better understand the manufacturing and quality issues associated with drug products containing nanomaterials. In particular, OTR is establishing clear standards to pave the way for the approval of future generics containing nanomaterials. Currently, studies are focused on identifying the critical processes and the material properties that can impact the quality of drug products containing nanomaterials. Figure 14 depicts some of the factors that could impact the quality of nanomaterial-based drug formulations within the context of efficacy and safety. Obviously, manufacturers should select and implement the right quality control measures so that any variability can be captured and accounted for.

To reduce variations in product quality, OTR encourages the use of advanced manufacturing techniques. In this context, OTR has been collaborating with scientists at the University of Connecticut (grant numbers HHSF223201310117C, HHSF223201610121C, and 1U01FD005773-01) to develop a platform for the continuous manufacturing of nanomaterials, which should allow better control over the manufacturing and quality of the process, and which should potentially lead to higher-quality products. In nanomaterial formulations, excipients play a more significant role than in traditional ones, but their characterization inside complex matrices has been only recently applied, and their critical attributes are not yet well understood. Regarding this, OTR collaborates with CDER’s Office of Generic Drugs to determine whether current characterization tools and standards for excipients are sufficient to support generic product development or whether different ones are needed. This research is funded in part by the Generic Drug User Fee Amendments (known as GDUFA II). Currently, very few nanomaterial-containing drug products have generic versions on the market. Additionally, to better evaluate the nanoproduct’s quality, safety, and efficacy, OTR research also focuses on determining, using advanced in vivo and in vitro analytical experiments, how the drug is released from nanocarriers and on establishing the relationship between in vivo and in vitro measurements.

### 5.2. Providing Nanotechnology Guidance and Information

The Nanotechnology Risk Assessment Working Group (NRAWG) is an organization that works to assess the potential impact of nanotechnology on pharmaceuticals. It aims at developing standards for nanomaterials used in drug development and at facilitating the advancement of nanotechnology. Promisingly, the working group established that, in most cases, current evaluation practices are adequate to evaluate medicines that include nanomaterials. On the other hand, CDER has worked over the past several years to understand the properties of nanomaterials when they are used in drug products to inform and ensure the development of a regulatory framework that can appropriately assess the impact of the unique physical properties of NPs on the safety and efficacy of nanomedicines. Recently, CDER issued a draft guidance for the industry titled “Drug Products, including Biological Products, that Contain Nanomaterials” [200]. CDER projects involved research on nanomaterial characterization and safety assessment in drug products, aimed at identifying the limitations of current test methods to assess the quality and safety of NP-based therapeutics and at evaluating the influence of nanotechnology application on the product characteristics, including stability and content uniformity. In this context, several peer-reviewed research articles that inform the scientific community on findings and advancements have been reported in the literature [201,202,203,204,205,206].

#### 5.2.1. Safety of Nanocarriers

For years, we have been witnessing the exponential growth of nanotechnology. In nanomedicine, NPs are used as pharmaceutical drug carriers with applications in both diagnostics and therapy. These NPs, including polymeric NPs, nanoemulsions, liposomes, and solid NPs, are suggested to have potential clinical applications [192]. However, their clinical applicability depends on different parameters, such as their physical and chemical properties, drug loading efficiency, drug release, and, most importantly, the low toxicity or nontoxicity of the carrier itself [207]. Nanocarriers have unique properties that are very different from those of small drug molecules, such as their nano dimensions and high surface-to-volume ratio and are able to efficiently infuse through the intestinal barrier to the circulation. Currently, the toxic impact of NP properties is not totally clear [208]. The actual safety of nanomaterial-based drug formulations should not be underestimated, and more studies focusing on the risks associated with the extensive use of NPs and nanotechnology are necessary. In fact, despite our increased exposure to NPs, information regarding NPs’ safety is limited, and research on safe NPs and/or on the safety of NPs lags behind that on the possible application of NPs [209]. As represented in Figure 15, illustrating the number of scientific papers published from the year 2000 to date (except for the current year, 2023) reporting on NPs and those on safe NPs, it is evident that, while the research on NPs is extensive (787,017 papers), that focused on the development of safe NPs, as well as studies concerning their toxicity, is dramatically limited (10,942 papers, 72-fold lower).

Despite the unequivocal advantage of using NPs for clinical application, some studies have suggested that NPs can be toxic. NPs could cause molecular toxicity, cell toxicity, tissue toxicity, or immunological toxicity. Exposure to NPs may be through lungs, injection, ingestion, or skin absorption, while the organ distribution includes the liver, spleen, and kidney. The brain is suggested as a potential target for NP distribution; however, direct evidence is still lacking [207]. NPs may enter cells by endocytosis and exert toxic effects, causing mitochondrial dysfunction, OS, inflammation, and DNA damage, both in animals and in humans [210,211]. The dimensions, modification of the surface, surface charge, composition, shape, and state of aggregation of NPs are pivotal factors in governing NP distribution in different organ systems following their exposure and in influencing their possible toxicity [207]. Increasing proof has demonstrated nanotoxicity caused by the induction of autophagy [212,213]. Additionally, the possible penetration of NPs into the skin, which could lead to damage to epidermal cells or their possible accumulation in secondary organs following biodistribution, is also of great concern [214,215]. These studies have demonstrated the ability of NPs to accumulate in cells and to induce organ-specific toxicity, and due to the ever-increasing human exposure to NPs, the design of progressively safer nanomaterials and the development of strict guidelines for their development with regard to toxicity testing are urgent [207]. On the other hand, there are also growing reports on the safety evaluation of nanocarriers, and there is a scenario of rising findings establishing that certain bioactive compounds loaded on nanocarriers are efficient and safe, thus being usable in medicine. In this regard, it has been demonstrated that after the ingestion of biodegradable polymers, chitosan–sodium alginate–oleic acid-based NPs loaded with lutein (LNCs) with a dose of 10 mg/kg body weight, no mortality and no morphological or clinical changes in rats were revealed [208]. Table 12 reports some examples of nanoformulations that have been demonstrated to be safe in in vivo or in vitro experiments.

However, while a vast array of nanocarriers are under development, many of which are undergoing advanced clinical trials, relatively few have achieved full translation to clinical practice. This slow uptake may be due, in part, to the need for a more rigorous demonstration of the safety of these new nanotechnologies. In the following, we provide a table (Table 13) in which the main studies on the acute and chronic toxicity of the most frequently used NPs are included.

As mentioned above, the toxic effects of NPs are dependent on their size, shape, chemical composition, extent of agglomeration, crystallinity, composition, and particle size distribution [207]. In addition, like all drugs, the dissolution rate of NPs can affect their acute toxicity. Unlike polymeric NPs or lipid-based NPs, inorganic NPs such as metal oxide NPs, including zinc, iron, and silver oxide NPs, are believed to dissolve after exposure, and it is supposed that the release of free ions associated with these inorganic NPs contributes to their toxicity. In this regard, studies have demonstrated that insoluble NPs of cerium oxide, titanium dioxide, and zirconium dioxide exhibited no toxicity when used in human mesothelioma cells at concentrations up to 30 µg/mL [222]. In contrast, soluble iron oxide and zinc oxide NPs were toxic at similar concentrations after three days of exposure [222]. This study exactly matches the results reported in several other works on the chronic toxicity of metal NPs, suggesting that the solubility of inorganic NPs is a key property in assessing their harmfulness due to ROS formation and the inactivation of enzymes [234,235]. A study focused on the evaluation of the toxicity of different metal oxide particles (CuO, TiO_2_, ZnO, CuZnFe_2_O_4_, Fe_3_O_4_, and Fe_2_O_3_) and that of carbon NPs and multiwalled carbon nanotubes (MWCNTs) demonstrated that when tested on the human lung epithelial cell line A549, cytotoxicity, DNA damage, and oxidative lesions were determined mainly for CuO NPs [223]. Importantly, the acute toxicity assessment of NPs is not sufficient to evaluate their safety, because exposure to NPs is a continuous daily process, such as the exposure of workers in the manufacturing sector or exposure through daily applied cosmetics. Further, the dissolution or degradation of NPs may take a significant amount of time, possibly much longer than the elimination of the therapeutic that they are carrying, while the products of degradation of NPs may themselves be toxic [207]. Finally, the biodistribution and accumulation of NPs may change over time [208]. Therefore, studies regarding the chronic exposure outcomes of NPs are necessary, which should include studies assessing the chronic use of NPs in humans in both clinical and industry settings and their bioaccumulation in the environment. Chronic exposure conditions need to be studied differently from acute exposure, since they could involve several steps that cannot be simulated in a single-step acute toxicological exposure [207]. Chronic studies of NP toxicity should try to cover all the toxicological observations (hematology, analysis of organs and tissues, genetic analysis). Further, to enhance the rigor of the data, chronic studies should use proper advanced analysis and a sufficient number of animals. In some cases, chronic studies should be performed over the lifetime of the animals (typically 2 years for rodents) [229]. In addition, in in vivo chronic toxicity studies, the route of exposure, dose, frequency, duration of exposure, and animal age and sex need to be considered, along with an understanding of the physicochemical properties of the NPs involved, including their composition, size, shape, charge, aggregation status, and degradation [207].

##### Titanium Dioxide: The European Case

Related to the initial work of CDER, which focused on evaluating the role of zinc oxide and titanium dioxide (TiO_2_) NPs in sun creams, studies were carried out to examine whether TiO_2_ NPs could penetrate normal skin [236]. An in vivo study using a pig model demonstrated that the TiO_2_ NPs did not penetrate the dermis [236]. Anyway, additional investigations carried out in the subsequent years have shown that TiO_2_ NPs can penetrate the protective barriers of the human body and accumulate in the liver, lungs, and digestive system with carcinogenic and genotoxic effects [237].

As a consequence, the use of TiO_2_ as a food coloring in food, food supplements, and feed was banned starting in 2020 in France and subsequently in all of Europe starting in mid-2022 based on a proposal from the European Commission approved on 8 October 2021 by the European Food Safety Authority (EFSA) committee [238]. Anyway, concerning current pharmaceutical products, TiO_2_ is still frequently used in solid oral pharmaceutical forms, such as tablets (coating), soft and hard capsules, pastes, and gels for oromucosal and sublingual use, as well as for semi-solid cutaneous and vaginal formulations [239].

At the request of the EU Commission, the European Medicine Agency (EMA) recently evaluated the possible impact of the removal of TiO_2_ from the list of authorized food additives for medicinal products. The final feedback from the EMA established that no single material had been identified that could provide the same combination of properties that are unique to TiO_2_ (opacity, enhanced contrast, protection from UV light, and finish/smoothness of the resulting product). Possible alternatives could be calcium carbonate, talc, and starch, but disadvantages have been identified with these alternatives [239].

Collectively, the feasibility of rapidly replacing TiO_2_ was not confirmed, mainly because each medicinal product containing TiO_2_ needs an individual review and assessment, which will require the investigation of alternatives, product reformulation, the generation of new data related to manufacturing, dissolution, stability, etc., and potential new clinical data, which will subsequently have to be assessed by the national authorities and EMA [239].

The direct and indirect impacts on medicines for human and veterinary use are expected to be aggravated in a scenario where Europe would be the only region globally to ban TiO_2_ as an excipient in medicines. The pharmaceutical industry should develop new drug formulations potentially for Europe only. Additionally, considering the scale of the use of TiO_2_, the time and costs involved in the reformulation, and the volume of products impacted, the replacement of TiO_2_ will almost certainly cause significant medicine shortages and the discontinuation of medicines on the EU/European Economic Area (EEA) market, with major implications for patients and animals. Collectively, it will not be possible to carry out the work for all products simultaneously, and the prioritization of product reformulations will be necessary. Taking into account the times estimated by pharmaceutical industries, the Quality Working Party (QWP) committee concluded that a transition period of 10 years or even longer would be required for the phasing out of TiO_2_ in medicines [239].

#### 5.2.2. Strategies to Reduce the Toxicity of Nanoparticles

It has been reported that NSs, NEs, and SLNPs are the less toxic ones being formulated using food-grade ingredients that have been generally recognized as safe (GRAS) by the FDA, such as lipids, proteins, polysaccharides, and surfactants [240]. Anyway, efforts are being made to limit the toxicity of nanomaterials. Table 14 collects the main strategies that have been developed to prepare safer NPs.

The most commonly used materials for coating NP surfaces include polyethylene glycol (PEG), polyvinylpyrrolidone (PVP), polyvinyl alcohol (PVA), poly(N–isopropylacrylamide) (PNIPAM), zwitterionic polymers such as poly(carboxybetaine) (PCB), poly(sulfo-betaine) (PSB), phosphorylcholine-based copolymers, and polysaccharides such as dextran and chitosan [253,254,255,256,257]. In this context, single-walled carbon nanotubes (SWCNTs) and MWCNTs may induce inflammation and fibrosis and promote cancer progression due to their surface chemistry, length, and aggregation state [258,259]. By applying the surface coating strategy using a nonionic triblock copolymer (PF108), Wang et al. improved their dispersion state and reduced their agglomeration, cellular uptake, and pro-fibrogenic effects [242]. In particular, in vitro experiments on bronchial epithelial BEAS-2B cells and phagocytic THP-1 cells and in vivo studies using mouse lungs demonstrated decreased toxicity due to a decrease in proinflammatory cytokines (IL-1), to low deposition in the lung, and to high protection against pulmonary fibrosis. In addition, Mutlu et al. demonstrated that CNTs coated with PF108 protected against lung toxicity and were cleared from the lungs after 90 days compared to non-coated CNTs, which aggregated and induced granulomatous lung inflammation and fibrosis [243]. More recently, Rosso et al., by coating previously synthetized ligand-free carbynoid-encapsulated gold nanocomposites (Au@Carbynoid NCs) with pluronic-F127 copolymer (PF127), achieved a fully biocompatible colloidal solution of an Au@Carbynoid/copolymer nanocomposite (NC), as confirmed by cytotoxicity studies on human skin fibroblasts [244]. Due to the stability of colloidal dispersions of Au@Carbynoid NCs functionalized with PF127 and their biocompatibility, green carbynoid-based NCs are promising as drug carriers for biological applications. A doping technique specifically used to reduce the toxicity of inorganic NPs alters the crystal structure of materials through the addition of impurities to improve chemical and physical properties [260,261,262]. Possible dopants include aluminum, titanium, and iron, while flame spray pyrolysis (FSP) is a well-established technique used in NP doping and uses a rapid combustion method. ZnO NPs have wide applications in cosmetics, such as sun creams and electronics, but ZnO-induced pulmonary inflammation has been reported in humans. George et al. synthesized Fe-doped ZnO NPs by FSP and assessed their cytotoxicity in vitro using RAW 264.7 and BEAS-2B mammalian cells. The results demonstrated decreased ZnO dissolution, which correlated with reduced in vitro cytotoxicity [263]. Subsequently, in vivo studies also showed the reduced toxicity of Fe-doped ZnO NPs in zebrafish embryos and rodent lungs [264]. Among surface modifications to reduce NP toxicity, the alteration of the charge density and hydrophobicity has been reported to improve the efficacy of some NPs in biomedical applications and their targeted drug delivery ability [207]. The alteration of the surface chemistry properties of NPs can be achieved through the covalent binding of functional groups, such as anionic, nonionic, and cationic groups, onto their surfaces. In this context, Li et al. synthesized and assessed the toxicity of CNTs functionalized with anionic, nonionic, and cationic surface groups in vitro and in vivo. CNTs with anionic groups (carboxylate and polyethylene glycol) displayed the lowest pro-fibrogenic effects and uptake in THP-1 and BEAS-2B cells [207]. The surfaces of iron oxide NPs, whose toxicity is attributed to the release of hydroxyl radicals, were modified by functionalization with organic compounds, such as aldehyde, carboxyl, and amino groups, thus resulting in their stabilization, decreased toxicity, and increased biological compatibility [249,250,251,252,253,254,255,256,257,258,259,260,261,262].

## 6. Phytochemical-Loaded Nanomedicines: Where Are We and Where Are We Going?

Table 15 and Table 16 below give us the current scenario of the actual clinical application of PHYs (Table 15) and of nanotechnological PHYs (Table 16) developed so far [97].

In total (non-formulated PHYs (Table 15) and nanoformulated ones (Table 16)), the number of types of clinically applied PHYs is 23, but only 5 (21.7%) are currently administered to humans in the form of NPs (camptothecin, curcumin, irinotecan, paclitaxel, and vincristine) (Figure 16).

All five of these are approved to counteract tumors, and while vincristine and irinotecan are administered only as NPs (Table 16), the others are also dispensed as non-nanotechnologically engineered drugs (Table 15 and Table 16). While for vincristine, curcumin, and irinotecan, only one nanoformulation exists, and irinotecan-based NPs and vincristine-based NPs are already approved and marketed, for camptothecin and paclitaxel, 5 and 13 different nanoformulations exist, respectively, and 3 out 13 (23.0%) paclitaxel-based nanoformulations are already marketed (Table 16, Figure 17), while camptothecin-based ones are still in clinical trials.

Curiously, among non-nanoformulated PHYs, only paclitaxel is currently clinically approved (Table 15). Figure 18 shows the clinical status of non-formulated (Figure 18a) and nanoformulated PHYs (Figure 18b).

Collectively, while only 1 out of 21 (4.8%) non-formulated PHYs is currently approved, the number of clinically approved PHY-based NPs is 5 out of 21 (23.8%).

Phase I and phase II clinical trials are currently ongoing to assess the feasibility of combination therapies using associations of curcumin with 5-fluoro uracile (phase I) and taxotene (phase II) to counteract multidrug-resistant metastatic colon cancer and resistant metastatic prostate cancer [265].

## 7. Conclusions

With this review, we offer the scientific community the complete scenario concerning the most relevant PHYs, their sources, and their pharmacological properties. By reporting several case studies, the main NS and NE techniques used in the past and recently developed to nanoformulate different types of PHYs have been reviewed. As an original contribution, most information provided in the present paper is organized into reader-friendly tables, is graphically presented, and has also been statistically analyzed. We have realized that the interest in the use of nanotechnology to solve the solubility and bioavailability issues of pharmacologically active molecules (phytocompounds in the present case) is growing so rapidly that the review of only two of the least exploited nanotechnological approaches among those that have been reported so far, that is, NSs and NEs, has led to a very notable and important work. We decided to deal with these nanoformulation techniques precisely because, in our opinion, they are underestimated, despite their great potential and limited toxicity due to the use of GRAS ingredients compared, for example, to the use of inorganic and organic NPs. Anyway, based on the number of case studies found in the literature, we conclude that, between NS and NE techniques, those based on NEs may be the most advantageous, as they are capable of obtaining nanoformulations made of smaller particles than NSs, thus providing nanomaterials with increased solubility, bioavailability, and pharmacological efficacy, as well as reduced toxicity. We also think that among the NE techniques reported here, those preparing SNEDDSs (including SMEDDSs, SNEDDSs, and SDEDDSs) are remarkably ingenious, as they lead to the obtainment of solid nanoformulations that are therefore more stable and that only once ingested, by exploiting biological fluids and peristaltic movements of the stomach, provide nanoemulsions that are readily and easily absorbable, avoiding phenomena of early degradation by enzymatic and microbial digestion occurring in the mouth. We are confident that the present review will inspire the interest of a large audience of experts in the sector of plant-derived products and nanomaterials and will encourage increasing research work to improve on the current results and to solve the pending issue regarding the possible toxic effects of NPs, thus increasing the number of PHY-based nanoformulations in clinical trials and then clinically approved products.

## Figures and Tables

**Figure 1 ijms-24-09824-f001:**
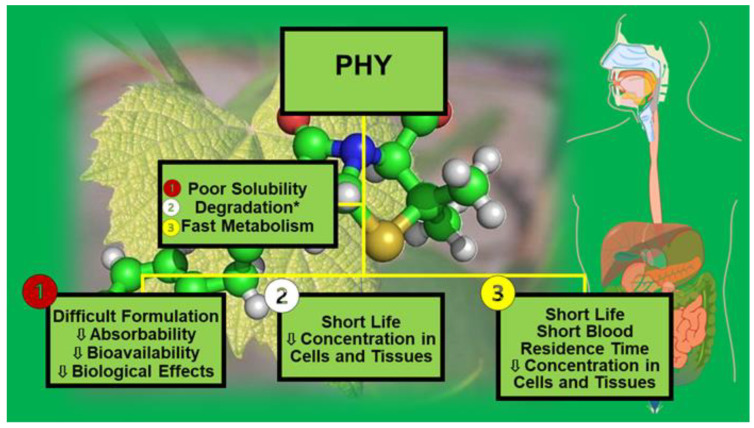
Main possible drawbacks related to most PHYs and events that limit their beneficial effects after oral administration. * Chemical, enzymatic, or microbial; ⇓ = reduced, decreased.

**Figure 2 ijms-24-09824-f002:**
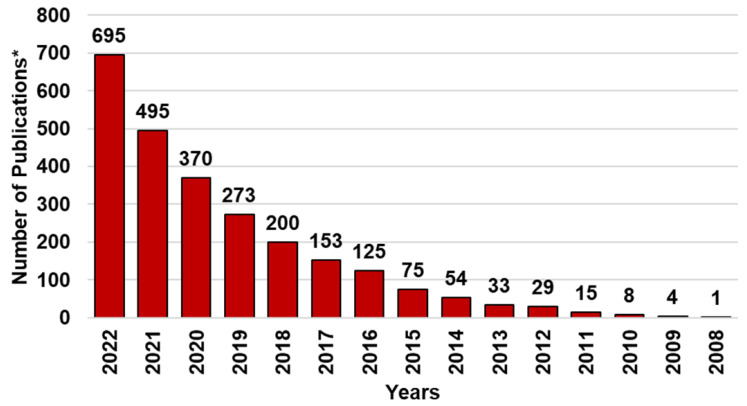
Number of publications per year over the last 15 years according to Scopus. * The research was carried out on 05 April 2023 using the following keywords: nanomaterials OR nanoparticles OR nanoformulation AND phytochemicals.

**Figure 3 ijms-24-09824-f003:**
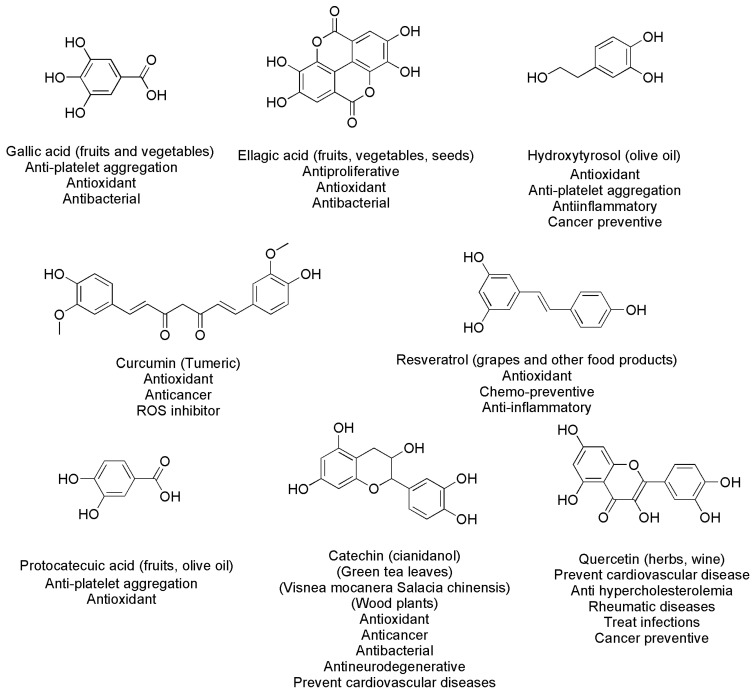
Structures and pharmacological properties of some plant-derived polyphenols. ROS = reactive oxygen species.

**Figure 4 ijms-24-09824-f004:**
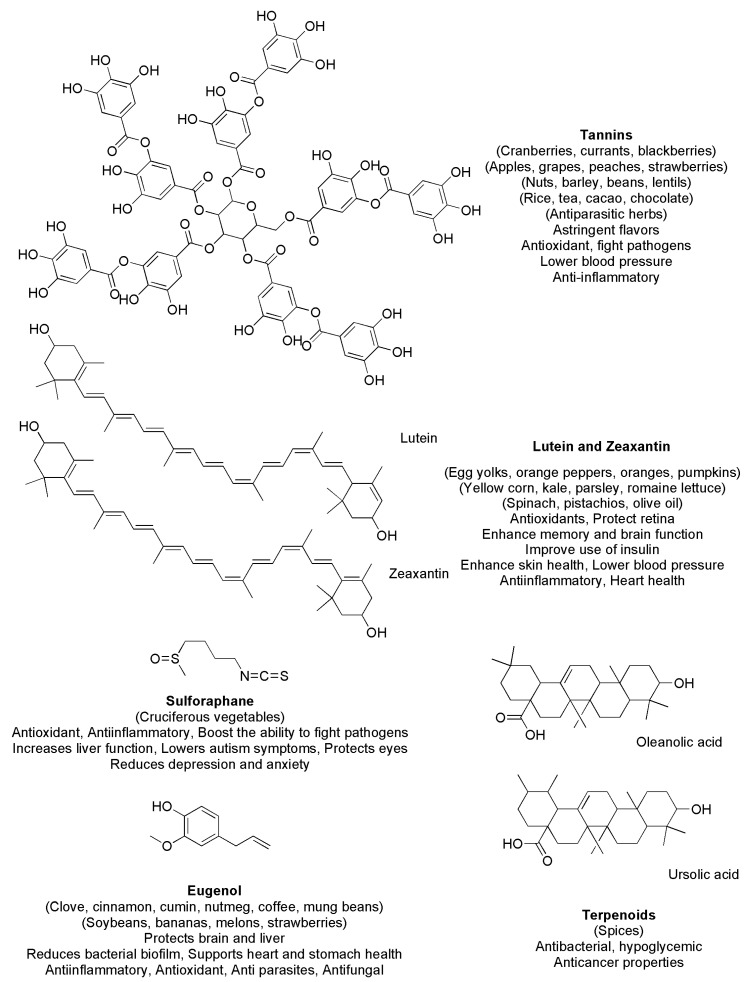
Chemical structures of some relevant PHYs.

**Figure 5 ijms-24-09824-f005:**
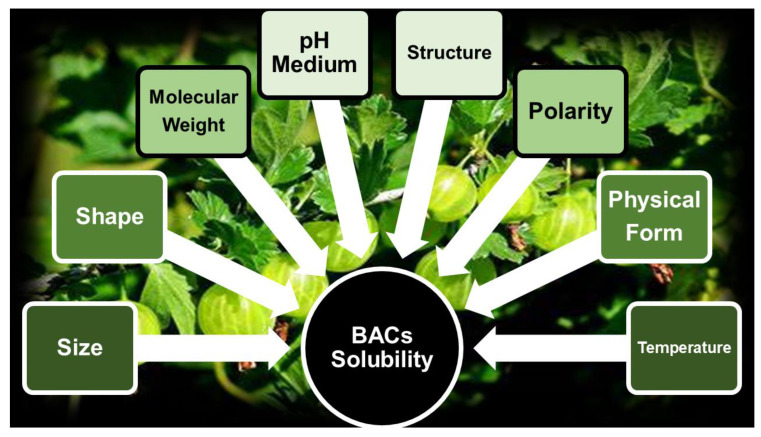
Main factors that can influence the water solubility of bioactive compounds (BACs).

**Figure 6 ijms-24-09824-f006:**
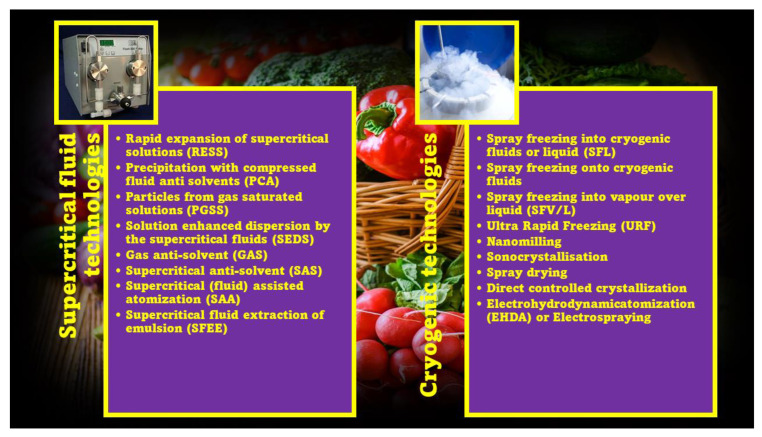
Supercritical fluid technologies and cryogenic technologies.

**Figure 7 ijms-24-09824-f007:**
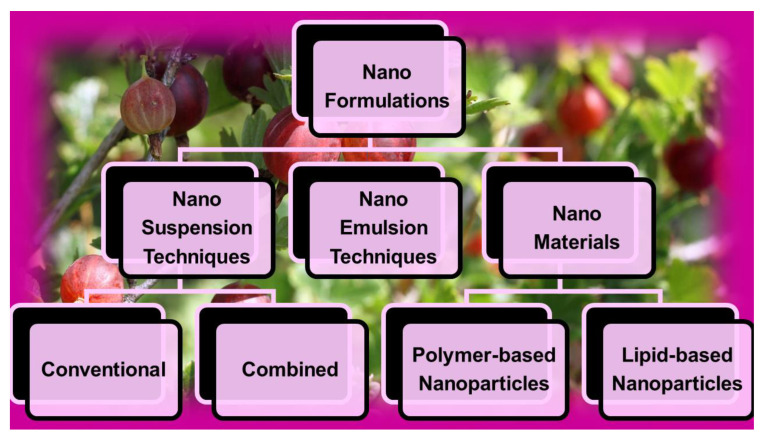
Main nanoformulation techniques.

**Figure 8 ijms-24-09824-f008:**
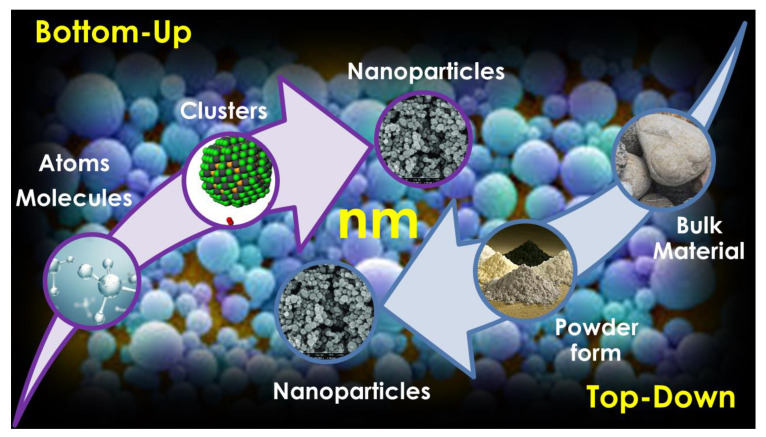
Bottom-up (B-U) and top-down (T-D) methods.

**Figure 9 ijms-24-09824-f009:**
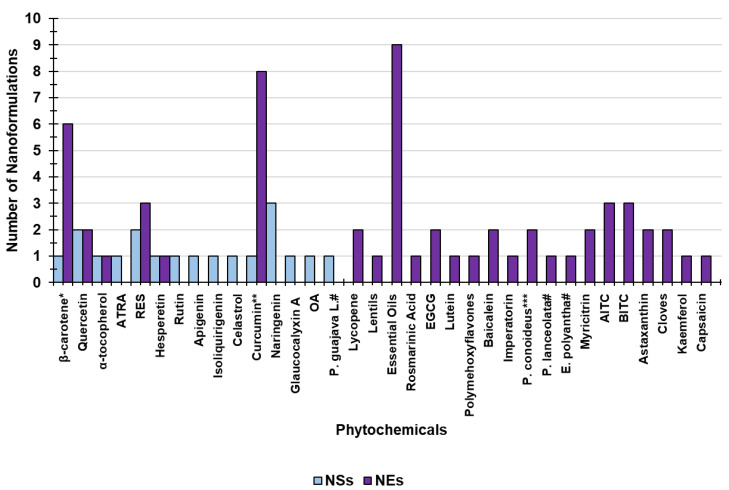
Main NS- and NE-based PHY formulations described in recent years. * Also including other carotenoids; ATRA = all-*trans* retinoic acids; RES = resveratrol; ** also including *Curcuma longa* extracts; OA = oleanolic acid; EGCG = epigallocatechin-3-gallate; *** red fruits; # plant extract; AITC = isothiocyanate; BITC = benzyl isothiocyanate.

**Figure 10 ijms-24-09824-f010:**
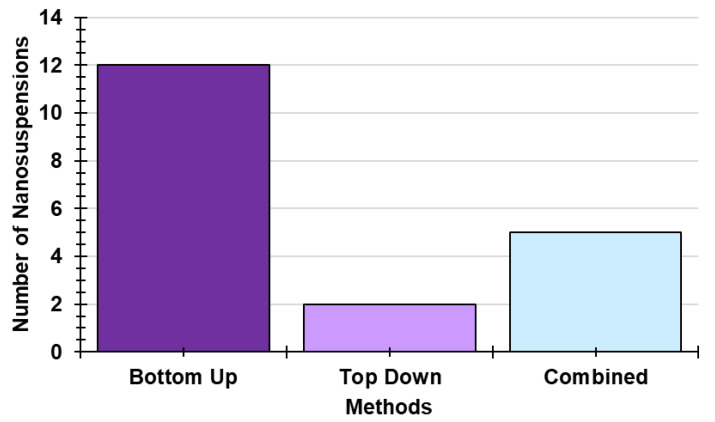
Main NS methods.

**Figure 11 ijms-24-09824-f011:**
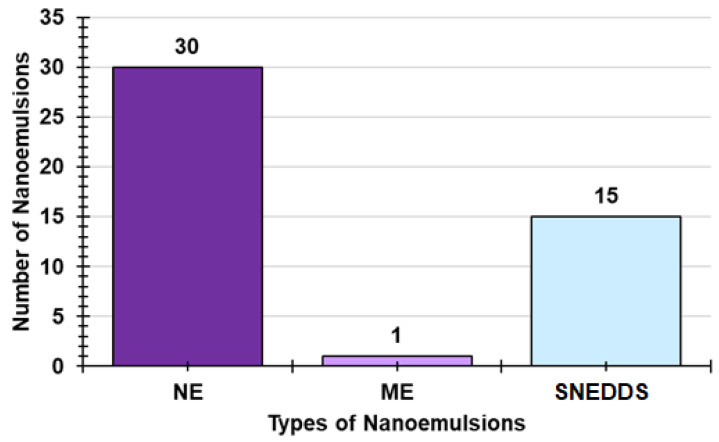
Main types of NEs.

**Figure 12 ijms-24-09824-f012:**
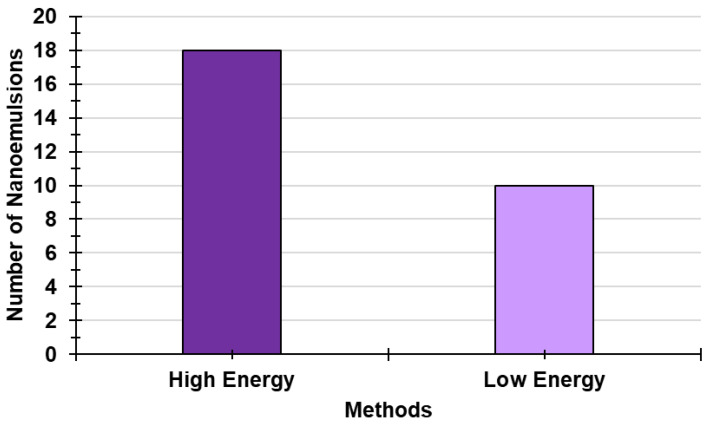
Main types of methods to prepare NEs.

**Figure 13 ijms-24-09824-f013:**
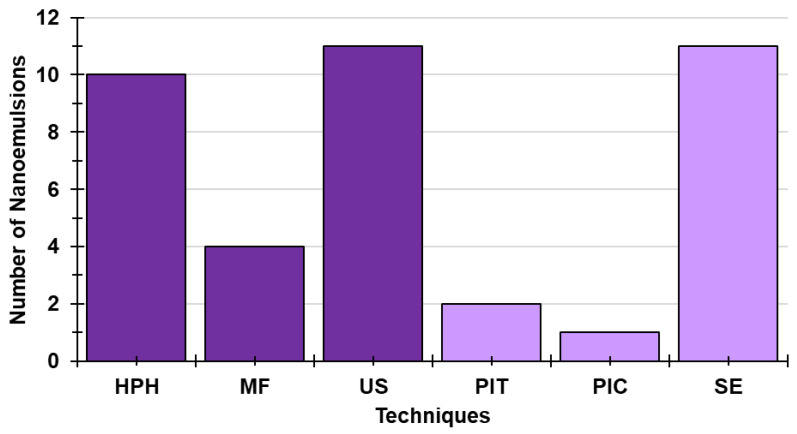
Main HETs (purple) and LETs (light purple).

**Figure 14 ijms-24-09824-f014:**
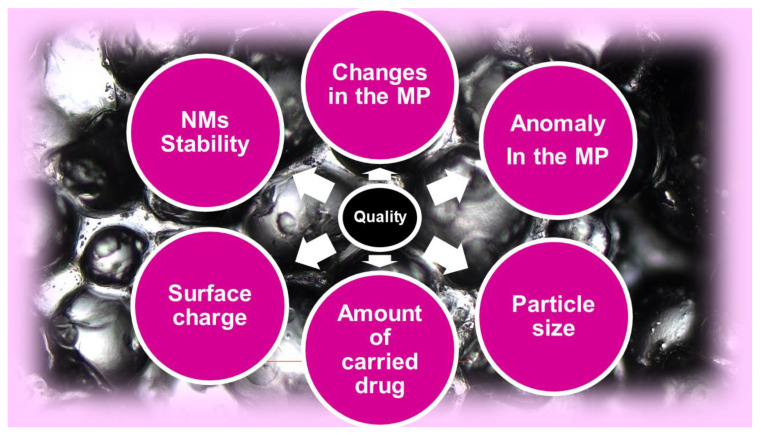
Main factors that could impact the quality of nanomaterial-based drug formulation within the context of efficacy and safety. NMs = nanomaterials; MP = manufacturing process.

**Figure 15 ijms-24-09824-f015:**
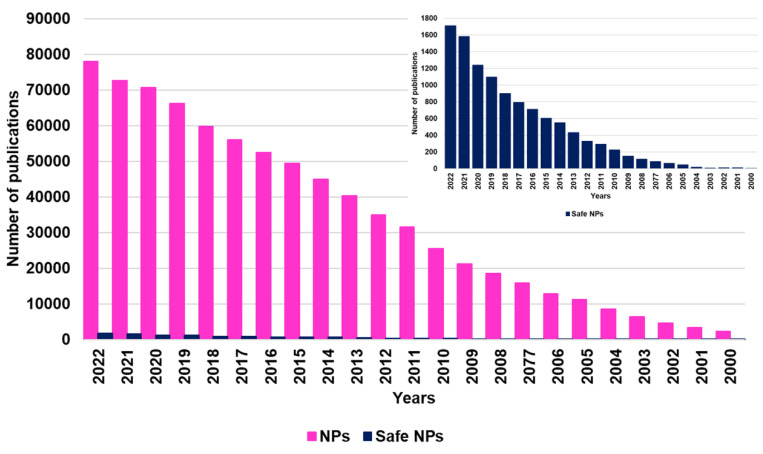
Articles identified in Scopus database from the year 2000 to 2022 using the key words nanoparticles (fuchsia) and safe nanoparticles (blue).

**Figure 16 ijms-24-09824-f016:**
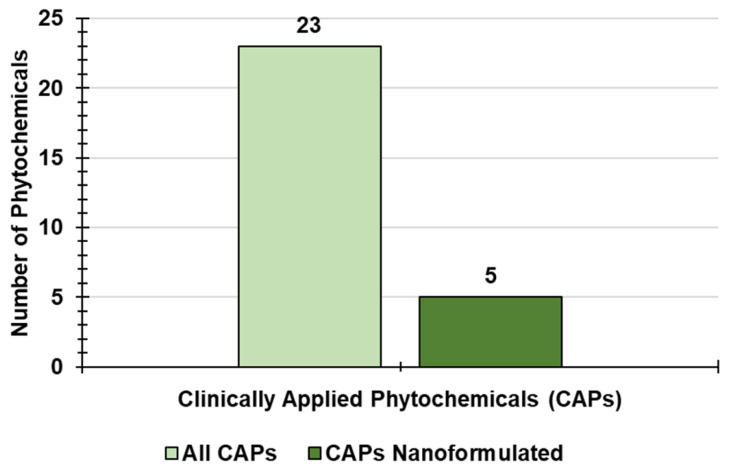
Clinically applied PHYs.

**Figure 17 ijms-24-09824-f017:**
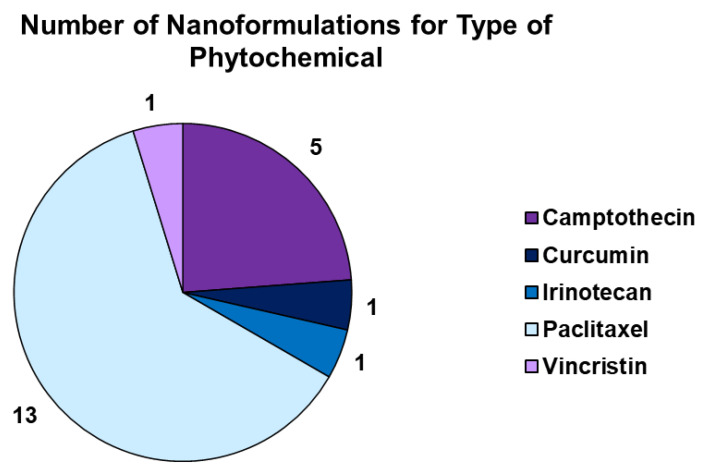
Number of nanoformulations existing for types of clinically applied PHYs.

**Figure 18 ijms-24-09824-f018:**
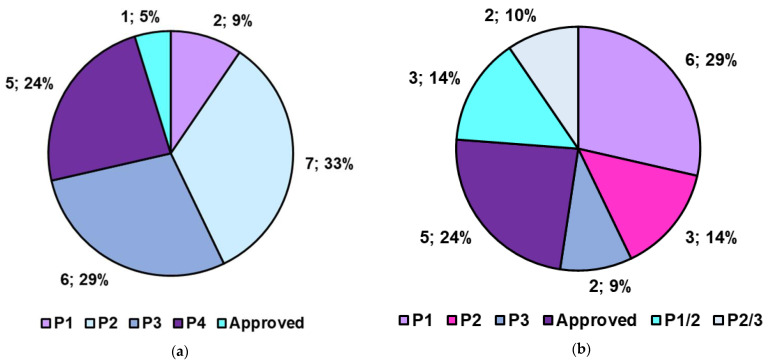
Clinical status of non-formulated PHYs (**a**) and of PHY-based nanoformulations (**b**).

**Table 1 ijms-24-09824-t001:** Sources of PHYs and their main benefits.

Edible Plants		Essential Oils	Benefits
Brightly Colored	Not Brightly Colored		
Berries	Potatoes Almonds Pecans Pistachios Cauliflower Walnuts Cashews Hazelnuts Tea Dark chocolate Cacao beans Barley Beans Lentils Rice Coffee Mung beans Soybeans Cloves Cinnamon Cumin Nutmeg	Pine needles Cedar Lavender	Boost the immune system [29] Combat OS and FR [30] ⇓ Blood sugar levels [30,31] ⇓ Blood pressure [29] ⇓ Diabetes risk [30,32] ⇓ Serious health issues [30,32] Prevent chronic disease [29,30] Protect from pathogens [33,34,35] Protect brain and liver [29] ⇓ Cholesterol [29] ⇓ Inflammation [30,36,37] Support detoxification [30,38] Ward off osteoporosis [29]
Cranberries			
Blackberries			
Strawberries			
Cherries			
Currants			
Grapes			
Plums			
Purple potatoes			
Red Cabbage			
Cabbage			
Kohlrabi			
Broccoli sprouts			
Apples			
Bananas			
Peaches			
Antiparasitic herbs			
Egg yolks			
Orange peppers			
Oranges			
Pumpkins			
Yellow corn			
Kale			
Parsley			
Romain lettuce			
Spinach			
Olive oil			
Melons			

OS = oxidative stress; FR = free radicals; ⇓ reduces, lowers.

**Table 2 ijms-24-09824-t002:** Most relevant types of PHYs, their sources, and the associated beneficial health effects.

PHYs	Sources	Health Benefits
Carotenoids [41,42]	Carrots, tomatoes, parsley Orange and green leafy vegetables Chenopods, fenugreek Spinach, cabbage, radish, turnips	Act as antioxidants Protect against uterine, prostate, colorectal, lung digestive tract cancers
Phytosterols [42,43]	Vegetables, nuts, fruits, seeds	Suppress the growth of diverse tumor cell lines
Limonoids [42]	Citrus fruits	Inhibit phase I enzymes and induce phase II detoxification enzymes in liver Provide protection to lung tissue, detoxify enzymes
Curcuminoids [42]	Turmeric, curry powder, mango, ginger	Analgesic, anti-inflammatory, anticancer, antioxidative Anti-depressive Protective against hay fever and depression, ⇓ cholesterol and itching risk
Indole compounds [42] (indole-3-carbinol)	Cabbage, cauliflower, broccoli, kale Brussels sprouts	Strong antioxidant, DNA protector, chemo-preventive, anticancer ⇑ Heart health
Alkaloids [44]	Plants (also animals and bacteria)	Antimalarial, antiasthma, anticancer, cholinomimetics Vasodilatory, antiarrhythmic Analgesic, antibacterial, antihyperglycemic, psychotropic, stimulant
Phytoprostanes [45] Phytofurans [45]	Almonds, vegetal oils, olives, algae Passion fruit, nut kernels, rice	Immunomodulators, anti-inflammatory, antitumor
Polyphenols [42]	Fruits, vegetables, cereals, beverages, legumes Chocolates, oilseeds	Action against free radicals, anti-inflammatory, anti-allergenic Inhibition of platelet aggregation, protective against hepatotoxins
Flavonoids * [42]	Fruits, vegetables, cereals, beverages, legumes Chocolates, oilseeds	Action against free radicals, anti-inflammatory, anti-allergenic Inhibition of platelet aggregation, protective against hepatotoxins
Iso-flavonoids ** [42]	Fruits, vegetables, cereals, beverages, legumes Chocolates, oilseeds	Action against free radicals, anti-inflammatory, anti-allergenic Inhibition of platelet aggregation, protective against hepatotoxins
Anthocyanidins ** [42,46,47,48] Anthocyanins ** [42,46,47,48]	Fruits, vegetables, cereals, beverages, legumes Chocolates, oilseeds	Act against free radicals, anti-inflammatory, anti-allergenic Inhibit platelet aggregation, protect against hepatotoxins Help control weight, prevent heart disease Increase insulin sensitivity Reduce inflammation, decrease diabetic complications, protect DNA Protect the brain Boost other PHYs and phytonutrients
Glucosinolates [42]	Cruciferous vegetables	Protection against cancer of colon, rectum, stomach
Phytoestrogens [42]	Legumes, berries, whole grains, cereals Red wine, peanuts, red grapes	Protection against bone loss, heart disease, cardiovascular diseases Protection against breast and uterine cancers
Terpenoids [42] Isoprenoids [42]	Mosses, liverworts, algae, lichens, mushrooms	Antimicrobial, antiparasitic, antiviral, antiallergic, anti-inflammatory Chemotherapeutic, antihyperglycemic, antispasmodic
Fibers [49]	Fruits and vegetables (green leafy), oats	⇓ Blood cholesterol, ⇓ cardiovascular disease
Polysaccharides [42]	Fruits and vegetables	Antimicrobial, antiparasitic, antiviral, antiallergic, anti-inflammatory ⇓ Serum, ⇑ defense mechanisms
Saponins [42]	Oats, leaves, flowers, green fruits of tomato	Protection against pathogens, antimicrobial, anti-inflammatory Antiulcer agent
Tannins [42]	Cranberries, currants, blackberries, apples, grapes, peaches Strawberries, almonds Hazelnuts, pecans, pistachios, walnuts, barley beans, lentils, rice Tea, cacao beans Dark chocolate, antiparasitic herbs	Act as antioxidants, fight pathogens, ⇓ blood pressure, ⇓ inflammation ⇓ Serious health risks Regulate the immune system
Lutein [42] Zeaxanthin [42]	Egg yolks, orange peppers, oranges, pumpkins, yellow corn, kale Parsley, romaine lettuce, spinach, pistachios, olive oil	Protect retina from damage, ⇑ eye function ⇑ Memory and brain function Promote the body’s use of insulin, ⇑ skin health, ⇓ blood pressure ⇓ Inflammation Support heart health
Eugenol [50]	Cloves, cinnamon, cumin, nutmeg, coffee Mung beans, soybeans, bananas, melons Strawberries, tomatoes	Acts as anti-inflammatory and antioxidant, eliminates parasites, fights fungi Inhibits serious health concerns, protects the brain and the liver, ⇓ Bacterial biofilm Supports heart and stomach health

⇑ Improved, higher; ⇓ reduced, lower; * a subgroup of polyphenols; ** a subgroup of flavonoids.

**Table 3 ijms-24-09824-t003:** General classification of the different color groups *.

Color Group	Foods	PHYs	Properties	Refs.
Green	Asparagus, avocados, celery, cucumbers Green beans, green peppers, kale, kiwi Spinach, zucchini	EGCG, glucosinolates Indoles, isoflavones Isothiocyanates, lutein, and zeaxanthin Sulforaphane	Promote wound healing and healthy gums Support arteries, blood cells, eyes, liver, and lungs	[18,66]
Purple	Black beans, blackberries Eggplants, elderberries, plums Purple cabbage, purple grapes, raisins	Anthocyanins, flavonoids, phenols Tannins, RES	Protect against serious health issues Support arteries, bones, brain, cognition, healthy aging, and heart	[18,66]
Red	Cherries, cranberries, kidney beans Red beans, strawberries, tomatoes Watermelon	Anthocyanins, ellagic acid, eugenol Hesperidin Lycopene, tannins, quercetin	Protect against heart disease and other serious health issues Support prostate, urinary tract, and DNA health	[18,66]
Yellow	Apricots, cantaloupe, carrots, grapefruit Yellow pears, yellow peppers Yellow winter squash	α-Carotene, β-carotene, β-cryptoxanthin, Lutein, zeaxanthin, hesperidin	Boost the immune system, support heart and vision health	[18,66]
White	Apples, cauliflower Great northern beans Mushrooms, onions	Allicin, ECGC, glucosinolates, indoles Tannins, quercetin	Protect against heart disease and other serious health issues Support arteries, bones, and circulation	[18]

* The present table was constructed by the authors using information found in the literature [18]; the row color reproduces the color of PHYs contained in the reported foods, which are responsible for the food’s coloration and appearance.

**Table 4 ijms-24-09824-t004:** Advantages and disadvantages of the most commonly used techniques and approaches for enhancing the solubility of bioactive compounds *.

Methods	Advantages	Disadvantages	Ref.
Emulsion solvent extraction Double-emulsion solvent extraction	Low cost	⇑ Residual solvent ⇓ Encapsulation efficiency (EE) Thermal degradation Multi-step processes Micronization step needed	[17]
SD	Improved bioavailability Taste masking Modified release achievable Aseptic manufacturing Fine powders Improved stability	High overhead Possible thermal degradation Not for every compound Not widely available Multi-step processes Micronization step needed	[17]
Liquid antisolvent technique	Low cost	⇑ Residual solvent ⇓ EE Thermal degradation Multi-step processes Micronization step needed	[17]
Spray freeze-drying (SFD) *	Higher rate of freezing Independent control over particle size	⇓ Biological activity Possible protein denaturation Excipients required	[17]
Spray freezing into liquid (SFL) *	Higher level of biological activity High degree of atomization Ultra-rapid freezing (URF) Formation of highly porous amorphous NPs	Possible ⇑ viscosity of the feed liquid Limited applications ⇑ Cost of equipment Time- and energy-intensive	[17]
Thin-film freezing (TFF) **	High-yield products Flexibility for processable drugs Large-scale production Simple, efficient, robust process ⇑ Stability of the protein product	⇑ Cost of equipment	[17]
Supercritical fluid extraction (SFE)	Single-step process Controllable particle size Controllable morphology Controllable crystallinity Monitorable residual solvent	⇑ Cost of equipment	[17]
Solvent evaporation technique	Low cost	⇑ Residual solvent ⇓ EE Thermal degradation Multi-step processes Micronization step needed	[17]

* The present table was created by the authors using information found in the literature [17]; ** cryogenic particle engineering methods [77]; ⇑ improved, higher; ⇓ reduced, lower.

**Table 5 ijms-24-09824-t005:** Physiological barriers to oral delivery.

Barrier	Region	Event	Factors	Drug Loss	Processes	Refs.
Biochemical	Stomach	Enzymatic degradation pH degradation ⇓ Interaction of drug with epithelial cells	HCl Pepsin * Mucus Lipases	94–98%	Deamidation Oxidation Hydrolysis	[80,81]
	Small intestine	Enzymatic degradation	Trypsins Chymotrypsins Carboxypeptidases Elastases			
	Colon	Long residence time (20 h)	⇓ Concentrations of digestive enzymes Neutral pH (6–6.7) ⇓ Fluid volumes/drug	N.A.	N.R.	[82]
Mucosal	Small intestine	⇓ Interaction of drug with epithelial cells ⇓ Penetration of drugs ⇓ Absorption	Two mucus ** layers with viscoelastic and hydrogel-like structure	N.R.	⇓ Transport ⇓ Absorption	[81,83]
		Microbial degradation	Gut Microbiota	N.R.	⇓ Transport ⇓ Absorption Degradation	[84]
Cellular	Intestinal epithelium ***	Governs the absorption of drugs	Epithelial ^1^ TJs ^2^ M-cells ^3^ Vitamin B12 ^4^ Hydrogen-coupled peptide transporters ^4^ Transferring receptors ^5^ IgG neonatal receptors ^5^ ABC	N.R.	⇓ Transport ⇓ Absorption Higher efflux	[81,85,86]
		Immunosuppression	Activated T-cell antibody- secreting plasma cells Macrophages Dendritic cells (DCs)	N.R.	Elimination	[84]
		Enzymatic degradation	Cytochrome P450 P-glycoprotein	N.R.	Oxidation	[79]

* Protein-digesting enzyme; ** secreted by goblet cells, with turnover rates of every 24–48 h and consisting of mucin, glycoproteins, proteins, carbohydrates, nucleic acids, lipids, salts, and antibodies; *** the outermost layer of cells exposed to luminal contents consisting of tight junction (TJ) cells, enterocytes, goblet cells, and microfold cells (M-cells); ^1^ responsible for drugs’ transcellular absorption; ^2^ responsible for paracellular drugs’ transport between adjacent epithelial cells; ^3^ responsible for lymphatic absorption; ^4^ responsible for drugs’ absorption by endocytosis; ^5^ other transferring systems; ABC = ATP-binding cassette family of efflux transporters; N.A. = not applicable; N.R. = not reported; ⇓ = reduced, decreased.

**Table 6 ijms-24-09824-t006:** Suspension-based nanoformulation techniques.

Nanosuspension Techniques		
Conventional Techniques		Combined Techniques (CTNIs)
Bottom-up (B-U)	Top-down (T-D)	Nanoedge™ Technique (Baxter Healthcare, Baxter Manufacturing, Firenze, Italy) H 69 Technology (SmartCrystal^®^ technology group, Pharmasol, Kaiserslautern, Germany) H 42 Technology (SmartCrystal^®^ technology, Pharmasol, Kaiserslautern, Germany) H 96 Technology (SmartCrystal^®^, Abbott/Soliqs, Ludwigshafen, Germany) Combination Technology (CTNO) (Eurofins CDMO Alphora Inc., Mississauga, ON, Canada)

**Table 7 ijms-24-09824-t007:** Examples of plant-derived bioactive compounds nanoformulated using conventional and combined nanosuspension techniques.

PHY	NPs/Mechanical Method (MM)/ Combined Techniques (CTNIs)	Particle Size	Characteristics
β-Carotene [109] (B-U)	*n*-Octenyl succinate-modified starch (NPs)	300–600 nm sized particles	⇑ Dispersibility ⇑ Coloring strength ⇑ Bioavailability
Quercetin [110] (B-U)	Maltodextrin (NPs)	753 nm sized particles	Water re-dispersible ⇑ RSA ⇑ ORAC
Quercetin [111] (T-D)	HPH (MM) + spray drying (MM)	400 nm sized particles	⇑ Antioxidant ⇑ Anti-inflammatory ⇑ Anticancer properties ⇑ Water solubility ⇑ Oral bioavailability
α-Tocopherol [112] (B-U)	Supercritical assisted process	150 nm sized particles	⇑ Solubility ⇑ Bioavailability
All-trans retinoic acid (ATRA) [113]	Nanoedge™ Technique	155 nm sized particles	Orally administrable 30′ Operation time
Resveratrol (RES) [114]	H 69 Technology	150 nm sized NPs	Orally administrable 10 cycles HPH/1200 bar
RES [17]	H 42 Technology	200 nm sized NPs	Orally administrable 1 HPH cycle at 1500 bar
Hesperidin [17] *	CTNO	599 nm sized NPs	⇑ Solubility Long-term stability Orally administrable Topically applicable Five homogenization cycles (1000 bar)
Rutin [17] **	CTNO	600 nm sized NPs	⇑ Solubility Orally administrable Topically applicable 1 Cycle of HPH at 100 bar
Apigenin [17]	CTNO	275 nm sized NPs	1 Cycle HPH at 300 bar
Isoliquiritigenin [115] T-D	Wet media milling (MM)	238 nm sized NPs (HPC-SSL) 354 nm sized NPs (PVP-K30)	⇑ Solubility ⇑ Cytotoxicity ⇑ Cellular uptake ⇑ Apoptosis induction ⇓ Toxicity
Celastrol [116] B-U	Antisolvent precipitation method	148 nm sized NPs	Stable in plasma Orally administrable ⇑ EE% ⇑ DL% ⇑ Solubility (vitro) ⇑ Cytotoxicity (vivo) ⇑ Cumulative release (48 h)
*Curcuma longa* L. extracts [117] B-U	Supercritical fluid expansion	47 nm sized NPs	⇑ Solubility
Naringenin (NRG) [118] B-U	Antisolvent sonoprecipitation	117 nm sized NPs	⇑ Absorption in GIT ⇑ Dissolution ⇑ Oral bioavailability
NRG [119] B-U	Precipitation–ultrasonication	118 nm sized NPs	⇑ Drug dissolution ⇑ Pharmacokinetic profile ⇑ Stability
NRG [120] B-U	HPH	81 nm sized NPs	⇑ Intracellular ROS level ⇑ Mitochondrial membrane potential ⇑ Caspase-3 activity ⇑ Lipid peroxidation status ⇓ GSH levels ⇑ Antitumor activity on DLA cells ⇑ Life span ⇓ Cancer cell number ⇓ Tumor weight
Glaucocalyxin A [121] B-U	Precipitation–ultrasonication	143 nm sized NPs	⇑ In vitro antitumor activity ⇑ In vivo anticancer efficacy
Oleanolic acid (OA) [122] B-U	Organic solvent evaporation	100 nm sized NPs	⇑ Stability ⇑ Saturation solubility ⇑ Dissolution rate ⇑ Cytotoxicity ⇑ Bio-efficacy ⇑ Bioavailability
*P. guajava* L. extracts [123] B-U	Nanoprecipitation	241–327 nm sized NPs	⇑ Antihyperglycemic activity ⇑ Physical parameters ⇑ Hepatic parameters ⇑ Renal parameters ⇑ Absorption ⇓ Metabolism ⇑ Stability
*Nigella sativa* L. [124] B-U	Nanoprecipitation	N.R.	⇑ Total phenolic content ⇑ Total flavonoid contents ⇑ Antioxidant activity ⇑ Antidiabetic activity ⇑ Antibiofilm activity ⇑ Bioavailability

RSA = radical scavenging activity; ORAC = oxygen radical absorbance capacity; * hesperidin nanocrystals can be found in the Platinum Rare cosmetic product (La Prairie, Volketswil, Switzerland); ** Rutin nanocrystals are in a cosmetic product launched by Juvena, St. Margrethen, Switzerland [17]; DL = drug-loading content; GSH = glutathione reductase; DLA = Dalton lymphoma ascites; ⇑ improved, higher, high; ⇓ decreased, lower.

**Table 8 ijms-24-09824-t008:** Main types of NEs and common oily phases.

Type of NEs		Oily Phase	Advantages	Drawbacks	Ref.
NEs o/w or w/o (100–500 nm)		Captex 355 Captex 8000 Witepsol Myritol 318 Isopropyl myristate Capryol 90 Sefsol-218 Triacetin Isopropyl myristate Castor oil Olive oil	PHY protection Controlled release Sustained release ⇑ DL%	No high-melting PHYs 5–10% additives * required	[1]
SEDDSs o/w **			⇑ Oral bioavailability Possibility of easy scale-up ⇑ DL% Allow delivery of peptides/lipids without the risk of digestion	5–10% additives * required For low-therapeutic-dose PHYs Many parameters The physicochemical properties of PHYs can influence the efficiency of oral absorption and the performances of SEDDSs	[1]
SNEDDSs ** (<50 nm)	SMEDDSs ** (100–200 nm)				
SDEDDSs ** (w/o/w) or (o/w/oil)					

SNEDDSs = self-nanoemulsifying drug delivery systems; SMEDDSs = self-microemulsifying drug delivery systems; SDEDDSs = self-double-emulsifying drug delivery systems; * surfactants, co-surfactants, and stabilizers; ⇑ high, higher; ** anhydrous systems.

**Table 9 ijms-24-09824-t009:** Main adopted high- and low-energy methods and their recent applications in nanoformulating PHYs.

Technique	Method	PHYs	Activity	Particle Size (nm)	Ref.
HET	High-pressure homogenization (HPH)	Jackfruit pulp extract * rich in carotenoids	Antioxidant	166	[131]
		Lycopene	Radical scavenging activity	92.6	[132]
		Lentil	⇓ Blood pressure ⇓ Bad cholesterol (LDL) ⇑ Good cholesterol (HDL) ⇓ Heart disease risk	149	[133]
	Microfluidization (MF)	β-Carotene	Antioxidant	140–160	[134,135]
	Ultrasonication (US)	Essential oils (EOs) (*Zataria multiflora*)	Antibacterial	91–211	[136]
		RES	⇓ Diseases by oxidative stress	20.4 **; 24.5 ***	[137]
LET	Phase inversion temperature (PIT)	Cinnamon EO	Antioxidant Antimicrobial	101	[138]
	Phase inversion composition (PIC)	Rosmarinic acid (RA)	Antiviral Anti-inflammatory Antioxidant	70–100	[139]
	Spontaneous emulsification (SE)	Key lime (*Citrus aurantifolia*) EO	Antibacterial	21	[140]
		Kaffir lime (*Citrus hystrix*) EO		28	
		Calamansi lime (*Citrofortunella microcarpa*) EO		60	

* Rich in carotenoids; ** NE; *** NE-cyclodextrin inclusion complex. ⇑ improved, higher, high; ⇓ decreased, lower.

**Table 10 ijms-24-09824-t010:** Emulsion-based formulations for delivering PHYs.

PHY	Emulsion Type/Method	Additives	Results	Ref.
Curcumin	NE	Tween 20	⇓ Toxicity ⇑ Bioavailability ⇑ Bioactivity ⇑ Anti-inflammatory	[151]
	o/w SNEDDS Mild agitation	Tween 80 PEG 600	⇑ Oral bioavailability ⇑ C max	[152]
	NE/HPH	PEG (3%)	⇑ Oral bioavailability ⇓ DHA levels	[153]
	Emulsion–diffusion evaporation	N.R.	⇓ Blood glucose levels ⇑ Insulin	[154]
	Interfacial prepolymer deposition and SE	Lipoid 100	Inhibition of OSCC cells ⇓ PI3K/Akt/mTOR ⇑ miR-199a	[155]
EGCG (E) + ALA (A)	SDEDDS	PGPR S721, P10, L23, and S40	⇑Photo-stability Antioxidant ⇑ EE	[156]
EGCG	NE (o/w)	BC WPI	No toxicity ⇑ Antioxidant	[157]
Carotenoids (*Paprika Oleoresin*)	SMEDDS	Tween 80	⇑ Solubility	[158]
Lutein	SMEDDS	Tween 80 Labrasol Transcutol HP/Lutro-E400 ^1^	⇑ Solubility ⇑ Bioavailability	[159,160]
Polymethoxyflavones (PMFs)	NE	Tween 20 Tween 85	⇑ Dissolution rate	[161]
*β*-Carotene	o/w NE	Tween 20	⇑ Emulsion stability ⇑ Solubility ⇑ Bio accessibility	[162]
Lycopene	Microemulsion (ME)	ESE 3GIO SML	⇑ Solubility	[163]
Quercetin	SNEDDS	Tween 80 PEG 400	⇑ Solubility	[164]
Naringenin Hesperetin	NE	Glycerin	⇑ Solubility ⇑ In vitro stability No cytotoxicity (in vitro) No hemolysis (in vivo) Anti-inflammatory	[165]
Baicalein	NE/HPH	PEGM Sodium oleate Hoechst 33,258 3,3-DODOXAP	⇑ Oral bioavailability ⇑ GIT permeability ⇑ Transcellular transport ability ⇓ Cytotoxicity	[166]
Imperatorin	NE/HSS NE/HPH	Polaxamer 188	⇑ Bioavailability Antiproliferative (MDA-MB-231)	[167]
*Pandanus conoideus* Lamk (red fruit)	SNEDDS	PPG Tween 20	⇑ Cytotoxic activity	[168]
*Pandanus conoideus* Lamk (red fruit)	NE/high-speed mixer	PPG SEPIGEL 305™	Antioxidant	[169]
*Plantago lanceolata* L.	SNEDDS	Labrasol or Kolliphor RH 40 Transcutol HP	⇑ Solubility ⇑ Permeability ⇑ Bioavailability ⇑ Pharmacological effects	[170]
Bay Leaves extracts (*Eugenia polyantha* Wight)		Tween 80 PEG 400		[171]
Myricitrin		Capryol 90 Cremophor RH 40 PEG 400 Cremophor EL Transcutol HP		[172]
Myricitrin		Cremophor EL35 Dimethyl carbinol		[173]
Quercetin		PEG 200 Tween 40 Tween 60 Tween 80 PEG 400 Transcutol HP		[174]
Baicalin		Peceol^®^ (14.3%, *w*/*w*) Kolliphor^®^ EL (57.1%, *w*/*w*) Transcutol^®^ P (28.6%, *w*/*w*)		[175]
AITC	Emulsion solvent evaporation	Polyvinyl alcohol (PVA) (3%)	⇓ Degradation ⇓ Volatility ⇑ Shelf life Sustained release ⇑ Toxicity to tumor	[176,177]
			⇑⇑ Anticancer activity ⇓ Toxicity	[178]
BITC	US	Tween 80 Decyl-β-d-glucopyranoside	⇑ EE%	[179]
BITC	US	Tween 80	⇑ EE% Good DL% MDA-MB-231 cell inhibition	[180]
	Heating stirring/sonication		Good long-term stability ⇑⇑⇑ EE Sustained release ⇑ Cytotoxicity (MDA-MB-231)	[181]
SEO	o/w NEs	Tween 20 Tween 80	⇑ Antibacterial effects	[182]
	o/w NEs Votexed/sonication	Tween 80	⇑ Antibacterial effects ⇓ Biofilm formation	[183]
RES	SNEDDS	Capryol 90 Cremophor EL Tween 20	⇑ Oral bioavailability ⇑ Anti-fatigue effect	[184]
	SMEDDS	Labrafil Labrasol RH40	⇑ Oral bioavailability	[185]
Astaxanthin + α-tocopherol	κ-Carrageenan o/w NE/SE	Span^®^ 80 (1%) PEG (1%) Tween^®^ 20 (1%)	⇓ Toxicity ⇓ Hyperglycemia ⇓ Diabetes complications ⇑ Wound healing	[186]
	κ-Carrageenan o/w NE Ultrasonication	None		
Cloves	SMEDDS	Tween 20 Tween 80	⇑ Anticancer effects ⇑ Antibacterial effects	[187]
Cloves	US	Tween 20 Tween 80 PEG	Cytoprotective effects ⇑ Anticancer effects ⇓ Toxicity	[188]
Kaempferol	HPH	Egg lecithin Tween 80	⇑ Permeation Safe Antioxidant capability ⇑ Drug delivery into rat’s brain ⇓ C6 cell viability	[189,190]
β-Carotene	o/w NE/US and MF	Casein	⇑ Water dispersibility ⇑ Chemical stability	[191]
	o/w NE/HPH	Porcine gelatin	⇑ Water dispersibility ⇑ Stability ⇑ Dispersibility in foods	[192]
Astaxanthin	o/w NE/SE	Lecithin	⇑ Stability ⇓ Photodegradation	[193]
Curcumin	o/w NE/HPH	SDS	⇑ Bioavailability ⇑ Antioxidant effects ⇑ Lipids digestion	[194]
	o/w NE/MF	Lecithin Tween 20 SMP	⇑ Stability ⇑ Antioxidant effects	[195]
	o/w NE/SE	Tween 80	⇑ Antimicrobial effects	[196]
Ginger EO	o/w NE/US	Tween 80	⇑ Antimicrobial effects ⇑ Antioxidant effects	[197]
Capsaicin	o/w NE/HPH+US	Tween 80	⇑ Antimicrobial effects ⇑ Physical properties	[198]

^1^ Co-surfactant; OSCC = oral squamous cell carcinoma; EGCG = epigallocatechin-3-gallate; ALA = α-lipoic acid; EE = entrapment efficiency; 3GIO = tri-glycerol monooleate; SML = sucrose monooleate; PEG = polyethylene glycol; PGPR = polyglycerol polyricinoleate; BC = bacterial cellulose; WPI = whey protein isolate; ESE = ethoxylated sorbitan esters; PEGM = poly(ethylene glycol) monooleate; 3,3-DODOXAP = 3,3-dioctadecyloxacarbocyanine perchlorate; DHA = docosahexaenoic acid; HSS = high-speed shearing; AITC = allyl isothiocyanate; BITC = benzyl isothiocyanate; SEO = *Satureja Montana* essential oil; PPG = polypropylene glycol; MDA-MB-231 = cell model of late-stage breast cancer; RES = resveratrol; SDS = sodium dodecyl sulfate; SMP = sucrose palmitate; ⇑ = increasing, improvement, higher, up-regulated; ⇓ reduced, lower. ⇑⇑ = highly improved; ⇑⇑⇑ = very very high.

**Table 11 ijms-24-09824-t011:** Post-administration events and related advantages concerning nanomaterial-based and non-nanomaterial-based drug formulations (NDF and NNDF).

Post-Administration Event	NNDF	NDF	Result	Advantages	Ref.
Interaction with specialized immune cells	NO	SI	Easier and faster macrophage-mediated targeting	⇓ Risk of side effects ⇓ Toxicity to nontarget organs ⇑ Effectiveness	[199]
Possible presence of a special coating	NO	SI	Capability to bypass immune cell attack		
		SI	Prolonged residence time in bloodstream		

⇑ improved, higher, high; ⇓ decreased, lower.

**Table 12 ijms-24-09824-t012:** Some examples of nanoformulations that were demonstrated to be safe in in vivo or in vitro experiments.

Formulation	Cargo	Administration Route	Animal Model Disease	Tests	Results	Ref.
SLN	Irinotecan	Rectal	Mice Cancer	Gel properties Pharmacokinetics Morphology Anticancer activity Immunohistopathology	Easily administered to the anus Rapid and strong gelling No damage to the rat rectum No body weight loss	[216]
Liposome	SP60015 (JNK inhibitor) Pitavastatin	Intravenous (iliac vein)	Male mice Aneurysm	Binding in vitro Binding in vivo Charging capacity Recharging capacity Drug release	Good drug transport Targeted drug release Repeatable drug release Safe	[217]
[P(bAsp-co-APIA)-PEG]	Docetaxel	N.A.	N.A. Cancer RA	DL% EE% Drug release	Biodegradable Biocompatible ⇓ Toxicity pH-sensitive	[218]
PS-NPs	*Dictyophora indusiata*	Gavage	Male mice IBD Colitis	Disease activity index Histological analysis Myeloperoxidase activity Goblet cells Mucous thickness Nitrogen oxide Cytokines Proteins	Effect against colitis Ameliorated intestinal injury ⇓ Oxidative stress ⇓ Proinflammatory cytokine ⇓ Inflammation ⇑ Mucins ⇑ Tight junction proteins (TJs) Restored intestinal microbiome ⇓ Harmful bacterial flora ⇑ Beneficial bacterial flora	[219]
[P(Asp-g-Im)-PEG]	Indole-3-acetic acid	Subcutaneous injection	Female nude mice Skin cancer	DL% EE% Morphology Drug release Cell viability Hemolysis Anticancer activity	⇓ Systemic toxicity (physiologic pH) ⇑ Antitumor efficacy ⇑ Accumulation in cancer cells ⇑ Release in cancer cells	[220]
mPEG-PCL micelles	Curcumin	Intravenous injection	Rats */Mice ** Breast cancer	DL% EE% Size Drug release Hemolysis In vivo organ toxicity In vivo anticancer effects	⇓ Systemic toxicity (physiologic pH) ⇑ Antitumor efficacy ⇑ Accumulation in cancer cells ⇑ Circulation time No mortality No organ toxicity No organ degeneration No necrosis No neutrophils No activation of immune response	[221]

SLNs = solid lipid NPs; JNK = c-Jun N-terminal kinase, a proinflammatory signaling molecule; [P(b-Asp-co-APIA)-PEG] = pH-sensitive poly{(benzyl-L-aspartate)-co-[N-(3-aminopropyl) imidazole-L-aspartamide]}-poly(ethylene glycol); IBD = inflammatory bowel disease; RA = rheumatoid arthritis; N.A. = not applicable; PS-NPs = polysaccharide-based nanocarriers; [P(Asp-g-Im)-PEG] = poly(aspartic acid-graft-imidazole)-poly(ethylene glycol); mPEG-PCL = mono methoxy poly (ethylene glycol)-poly (e-caprolactone) di-block copolymers; * mortality and in vivo toxicity (kidney, liver, heart, spleen); ** antitumor activity; ⇓ = reduced, decreased, lower; ⇑ = increased, improved, higher.

**Table 13 ijms-24-09824-t013:** Acute and chronic toxicity of NPs.

Acute Toxicity			
NP Type	Animal/Cells	Toxicity	Ref.
Fe_2_O_3_ NPs ** ZnO NPs **	Human mesothelioma cells Rodent mesothelioma cells	⇓ Overall cell culture activity ⇓⇓⇓ DNA content	[222]
CuO, TiO_2_ ZnO, CuZnFe_2_O_4_	A549 cells	Cytotoxicity DNA damage ⇑ OS by ROS Oxidative lesions	[223]
Nano-C60 fullerene aggregate	Human dermal fibroblasts Human liver carcinoma cells (HepG2) Neuronal human astrocytes	⇓ Normal cellular function Lipid peroxidation ⇑ ROS Membrane damage	[224]
TiO_2_ NPs	Brain microglia (BV2)	⇑ ROS Neurotoxicity	[225]
**Chronic toxicity**			
**NP Type**	**Animal/Cells**	**Toxicity**	**Ref.**
CNTs	Mice *	Asbestos-like pathogenicity	[226]
			[227]
MWCNTs	Female mice	Breast cancer metastasis	[228]
Inorganic NPs	Rodents Non-rodents	Genotoxicity Carcinogenesis Embryotoxicity	[229]
Al_2_O_3_ NPs ZnO NPs	Rats#	Hepato-renal toxicities ⇓ Hepatic expression of mtTFA and PGC-1α proteins	[230]
Au NPs	*Daphnia magna*	Mortality ⇓ Reproductive development ⇓ Reproductive fitness ⇓ Total eggs and offspring Aborted eggs	[231]
ZnO NPs	*Mytilus galloprovincialis*	⇓ Transcription of key genes involved in DNA damage/repair, antioxidation, and apoptosis	[232]
TiO_2_ NPs	Rats	Inflammation Lung injury ⇓ Alveolar macrophage function	[233]
ZnO NPs ** ZnCl_2_ NPs **	*P. subcapitata*	Cytotoxicity	[234]
ZnO NPs	RAW 264.7 cells BEAS-2B cells	Cytotoxicity OS	[235]

** Soluble NPs; CNTs = carbon nanotubes; * abdominal cavity; OS = oxidative stress; MWCNTs = multiwalled CNTs; # oral administration; ⇓ = reduced, decreased, lower; ⇑ = increased, improved, higher; ⇓⇓⇓ = strongly reduced, decreased, lower.

**Table 14 ijms-24-09824-t014:** The main strategies developed to prepare safer NPs.

Strategy	Features	Type of NPs	Results	Ref.
Use of next generation lipids *	⇑ High potency Biodegradability	SLNPs	Rapid elimination from plasma ⇑ Tolerability in preclinical studies ⇑ In vivo potency	[241]
Surface coating strategies *	Biocompatibility ⇑ Colloidal stability ⇓ Degradation Faster excretion ⇓ Accumulation Reversible coating Altered dispersion state	Polymeric SLNPs Inorganic NPs	⇑ Dispersion state ⇓ Agglomeration ⇓ Cellular uptake ⇓ Pro-fibrogenic effects	[242]
			⇓ Lung toxicity	[243]
		AuC/PF127 NCs	⇑ Stability ⇑ Biocompatibility Photodynamic therapeutic	[244]
Doping	Altered density of surface reactive chemicals ⇓ Binding energy of metal ions to oxygen ⇓ NP dissolution ⇓ Toxic ion release ⇓ ROS generation	Inorganic NPs	⇓ Dissolution ⇓ Toxicity	[245]
			⇓ Dissolution ⇓ Toxicity	[246]
Surface chemistry properties modifications	Altered charge density Altered hydrophobicity	CNTs	⇓ Pro-fibrogenic effects ⇓ Uptake in THP-1 and BEAS-2B cells	[247]
		AuNPs	⇓ Toxicity ⇓ Uptake in cells	[248]
		Fe_2_O_3_ NPs	⇑ Stability ⇓ Toxicity ⇑ Biocompatibility	[249,250,251,252]

* The surface of NPs can be covered with various substances, such as polymers, in single or multiple layers that can be either complete or incomplete; ⇓ reduced, decreased; ⇑ increased, higher, improved, high; AuC/PF127 NCs = carbynoid-encapsulated gold nanocomposites (NCs) functionalized with pluronic-F127 (PF127).

**Table 15 ijms-24-09824-t015:** Clinical status and biological activity of PHYs currently applied to humans.

Compound	Status *	Source	Activity
Artemisinin	Phase III	Artemisia annua	Anticancer
Ursolic acid	Phase II	Fruits (waxes of apples, pears)	Antioxidant
Thymoquinone	Phase II	Herbs Spices	Hepatoprotective Antioxidant Anticancer
Sulforaphane	Phase II	Brassica vegetables	Anticancer Antioxidant Antimicrobial Anti-inflammatory
PEITC	Phase II	Watercress	Anticancer (lung, oral)
Not specified ITC	Phase I	Cruciferous vegetables	Bladder cancer
Sinomenine	Phase III	Roots of *Sinomenium acutum*	Anti-inflammatory Anti-rheumatic
Silibinin	Phase IV	Milk thistle Coffee	Hepatoprotective Anticancer
Catechin	Phase IV	Green tea Beans	Antioxidant
Salvianolic acid B	Phase II	Red sage	Antioxidant Angiogenetic
RES	Phase IV	Grapes Blueberries Raspberries Mulberries	Antioxidant Anti-inflammatory Cardioprotective Anti-carcinogenic
Quercetin	Phase III	Fruits Red onions Kale	Anti-inflammatory Anticancer
Paclitaxel	Approved (Taxol^®^) FDA (1998)	Bark of Pacific yew tree	Mitotic inhibitor in cancer Chemotherapy
Genistein	Phase III	Plants (lupins, fava beans, soybeans)	Anticancer Anti-inflammatory
Lycopene	Phase IV	Tomato	Antioxidant Anticancer
EGCG	Phase III	Green tea White tea Black tea	Antioxidant Chemo-preventive
Epicatechin	Phase II	Woody plants	Antioxidant
Caffeic acid	Phase III	Coffee Eucalyptus	Anticancer Antioxidant Anti-inflammatory
Camptothecin	Phase I	Stem wood of the Chinese tree *Camptotheca acuminate*	Anticancer
Combretastatin	Phase II	Bark of *Combretum caffrum*	Anticancer
Curcumin	Phase IV	Tumeric	Inhibition of tumor cell proliferation Anti-inflammatory

The present table was created by the authors based on the example found in the relevant paper by Han et al. [97]; * clinical status was obtained from http://clinicaltrials.gov (top clinical status in drugs of intervention category). Accessed on 6 April 2023.

**Table 16 ijms-24-09824-t016:** PHY-based nanoformulations currently applied to humans.

Compound	Status *	Formulation Type	Indication
Camptothecin	Phase II	PEG–drug conjugate	Gastric cancer
	PhaseI/II	Polyglutamic acid–drug conjugate	Colon cancer Ovarian cancer
	Phase I/II	Cyclodextrin NP	Solid tumors, renal cell carcinoma, rectal cancer, non-small-cell lung cancer
	Phase I	HPMA–drug conjugate	Solid tumors
	Phase I	Fleximer–drug conjugate	Gastric cancer Lung cancer
Curcumin	Phase I	Liposome	Advanced cancer
Irinotecan	Onivyde^®^ Approved: FDA (2015)	Liposome	Metastatic pancreatic cancer
Paclitaxel	Abraxane^®^ Approved: FDA (2005, 2012, 2013)	NPs, albumin-bound	Breast cancer Non-small-cell lung cancer Pancreatic cancer
	Phase I	Polymeric micelle	Ovarian cancer
	Phase I	Polymeric NPs	Peritoneal neoplasms
	Phase I/II	Liposome	Ovarian cancer Breast cancer Lung cancer
	Phase II	Liposome	Triple-negative breast cancer
	Phase II	Liposome	Solid tumors Gastric cancer Metastatic breast cancer
	Phase II/III	PEG-PAA polymeric micelle	Gastric cancer Breast cancer
	Phase II/III	DHA–drug conjugate	Melanoma Liver cancer Adenocarcinoma Kidney cancer Non-small-cell lung cancer
	Phase III	Polyglutamic acid–drug conjugate	Lung cancer Ovarian cancer
	Phase III	Polymeric micelle	Advanced breast cancer
	Apealea^®^ Approved: EMA (2018)	Micelle	Ovarian cancer Primary peritoneal cancer
	Genexol-PM^®^ Approved: marketed in Korea and Europe (2007)	PEG-PLA polymeric micelle	Breast cancer Lung cancer
	Phase I	HPMA–drug conjugate	Solid tumor
Vincristine	Marqibo^®^ Approved: FDA (2012)	Liposome	Acute lymphoid leukemia

The present table was created by the authors based on the example found in the relevant paper by Han et al. [97]; * clinical status was obtained from http://clinicaltrials.gov (top clinical status in drugs of intervention category). Accessed on 6 April 2023.

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
