# Peer review of "Development of Phytochemical Delivery Systems by Nano-Suspension and Nano-Emulsion Techniques"

_ijms, 2023, doi:10.3390/ijms24129824_

Round 1
Reviewer 1 Report
The manuscript discusses nanosuspension and nanoemulsion based methods for formulating phytochemicals in more bioavailable nanoparticles. While I can see some useful scientific background in the manuscript, I am concerned that these "nanosuspension and nanoemulsion based methods" are not properly described. I expected to see more information about these interesting methods and their application in the review. Exactly the same text can be found on the Internet doi doi:10.20944/preprints202305.0658.v1. Thus the paper not considered appropriate at the present stage for publication, but should be subjected to a major revision.
Additional issues that should be addressed:
Major:
1. The introduction should be shortened.
2. Remove Methodology for Literature Search and Study Selection
3. The description of Phytochemicals (PHYs) to be shortened, it occupies a space from pp 3 to 15 and does not correspond to the main purpose of the review
4. The Nanotechnology section should be rewritten with a detailed description of nanosuspension and nanoemulsion based methods because this is the main purpose of the review
Author Response
The manuscript discusses nanosuspension and nanoemulsion based methods for formulating phytochemicals in more bioavailable nanoparticles. While I can see some useful scientific background in the manuscript, I am concerned that these "nanosuspension and nanoemulsion based methods" are not properly described. I expected to see more information about these interesting methods and their application in the review. Exactly the same text can be found on the Internet doi doi:10.20944/preprints202305.0658.v1. Thus the paper not considered appropriate at the present stage for publication, but should be subjected to a major revision.
We thank a lot the Reviewer for his suggestion, which enabled us to describe more extensively the NSs and Nes-based methods as specified later. For the rest, we make kindly note to the Reviewer that the text available online at doi:10.20944/preprints202305.0658.v1, it is not a duplicate of the present one, but it is the preprint of the manuscript that he reviewed. Preprint is a service offered by MDPI journals, to promote the article before publication.
Additional issues that should be addressed:
Major:
- The introduction should be shortened.
As asked, the Introduction section has been reduced. Please see at lines 50-53, 60-65, 88-93, and 112-114.
- Remove Methodology for Literature Search and Study Selection.
As requested, the paragraph “Methodology for Literature Search” has been removed. Please see at lines 138-148.
- The description of Phytochemicals (PHYs) to be shortened, it occupies a space from pp 3 to 15 and does not correspond to the main purpose of the review.
As asked, the description of PHYs has been reduced. Please see at lines 156-158, 160-163, 168-169, 175-177, 182-183, 189-194, 196-201, 216-251, 267-320.
- The Nanotechnology section should be rewritten with a detailed description of nanosuspension and nanoemulsion based methods because this is the main purpose of the review.
We thank the Reviewer for his suggestion, again. As asked, the nanotechnology section has been extensively improved by including a detailed description of NSs and Nes methods. Please, see at lines 548-611 and 754-824.
Reviewer 2 Report
1. Recently, several review papers have been published in MDPI. Please search the reference and include these useful references in this paper (e.g., Pharmaceutics 2023, 15(3), 889).
2. I believe this paper also introduce other administration routes besides oral. Please at least give some explanation on this point to make a smooth story.
3. So many abbreviations are used. It is better to summarize them.
4. Please introduce the barriers of oral delivery by nanotechnology. And also please mention that how the listed examples can overcome these barriers.
5. It is better to emphasize that nanotechnology can advance the function of drug delivery (https://doi.org/10.1021/jacs.0c09029).
6. Colchicine is another interesting phytochemicals having wide medical application. Recently, it also receive attention in COVID-19. Nanotechnology also has been reported to enhance the delivery of Colchicine. This paper might be useful (https://doi.org/10.1002/adma.202105254).
Author Response
Recently, several review papers have been published in MDPI. Please search the reference and include these useful references in this paper (e.g., Pharmaceutics 2023, 15(3), 889).
We thank the Reviewer for his useful suggestion. The indicated reference has been included in the Introduction Section. Please, see at lines 99-103.
- I believe this paper also introduce other administration routes besides oral. Please at least give some explanation on this point to make a smooth story.
We agree with the Reviewer. In effect, although the original title and most part of our paper focus mainly on the phytochemicals loaded NSs and NEs orally administrable, other administration routes are also introduced. In this regard, we have changed the title and consequently some parts of the manuscript in more generic forms, and as asked more details on the question have been included along whole paper. The Reviewer can find some examples in lines 16-19, 25-26, 70-72, 75-77, 82, 127-129, etc.
- So many abbreviations are used. It is better to summarize them.
The Reviewer is right, but on previous experience, lists of abbreviations are not allowed in the MDPI journals. Anyway, to address the Reviewer request, we have created a List of Abbreviations that has been included in Appendix A, at the end of the manuscript before the references list (line 1459).
- Please introduce the barriers of oral delivery by nanotechnology. And also please mention that how the listed examples can overcome these barriers.
As asked, the physiological barriers to oral delivery have been included in the new Section 4.1 and Table 5 (lines 415-432). The mode by which the listed examples overcome these barriers, when possible, was already included for each example in the original manuscript. Anyway, to meet the request of the Reviewer, general information on how nanosuspension and nano-emulsion-based formulations can overcome the physiological barriers to oral delivery have been inserted. Please, see at lines 495-499 and 702-717.
- It is better to emphasize that nanotechnology can advance the function of drug delivery (https://doi.org/10.1021/jacs.0c09029).
The request of Reviewer has been addressed. Please, see lines 405-414.
- Colchicine is another interesting phytochemicals having wide medical application. Recently, it also receive attention in COVID-19. Nanotechnology also has been reported to enhance the delivery of Colchicine. This paper might be useful (https://doi.org/10.1002/adma.202105254).
We thank the Reviewer for his suggestion that we have addressed. Pleas, see lines 103-109.
Round 2
Reviewer 1 Report
The authors have significantly improved the article. Now it can be published.
Author Response
We thank a lot the Reviewer for his/her appreciation.
Reviewer 2 Report
The authors addressed concerns well. I found now the paper very instructive and balanced, and I enjoyed reading it.
Author Response
We are very thankful to the Reviewer for his/her comment.